# Theoretical Limitations of Ensembles in the Age of Overparameterization

**Niclas Dern** [* 1]   **John P. Cunningham** [2]   **Geoff Pleiss** [3 4]

## Abstract

Classic ensembles generalize better than any single component model. In contrast, recent empirical studies find that modern ensembles of (overparameterized) neural networks may not provide any inherent generalization advantage over single but larger neural networks. This paper clarifies how modern overparameterized ensembles differ from their classic underparameterized counterparts, using ensembles of random feature (RF) regressors as a basis for developing theory. In contrast to the underparameterized regime, where ensembling typically induces regularization and increases generalization, we prove with minimal assumptions that infinite ensembles of overparameterized RF regressors become pointwise equivalent to (single) infinite-width RF regressors, and finite width ensembles rapidly converge to single models with the same parameter budget. These results, which are exact for ridgeless models and approximate for small ridge penalties, imply that overparameterized ensembles and single large models exhibit nearly identical generalization. We further characterize the predictive variance amongst ensemble members, demonstrating that it quantifies the expected effects of increasing capacity rather than capturing any conventional notion of uncertainty. Our results challenge common assumptions about the advantages of ensembles in overparameterized settings, prompting a reconsideration of how well intuitions from underparameterized ensembles transfer to deep ensembles and the overparameterized regime.

---
[*]Work done while at the Vector Institute. [1]School of Computation, Information and Technology, Technical University of Munich, Munich, Germany [2]Department of Statistics, Columbia University, Zuckerman Institute, New York, USA [3]Department of Statistics, University of British Columbia, Vancouver, Canada [4]Vector Institute, Toronto, Canada. Correspondence to: Niclas Dern <niclas.dern@gmail.com>.

*Proceedings of the 42$^{nd}$ International Conference on Machine Learning*, Vancouver, Canada. PMLR 267, 2025. Copyright 2025 by the author(s).

## 1. Introduction

Historically, most machine learning ensembles aggregated component models that are simple by today's standards (e.g. Hansen & Salamon, 1990; Opitz & Maclin, 1999; Dietterich, 2000). Common techniques like bagging (Breiman, 1996), feature selection (Breiman, 2001), random projections (Kabán, 2014; Thanei et al., 2017), and boosting (Freund, 1995; Chen & Guestrin, 2016) were developed and analyzed assuming decision trees, least-squares regressors, and other *underparameterized* component models incapable of achieving zero training error.

Researchers and practitioners have now turned to ensembles of *overparameterized* models, such as neural networks, which have capacity to memorize entire training datasets. Motivated by heuristics from classic ensembles (Mentch & Hooker, 2016), some have argued that ensembles provide robustness to dataset shift (Lee et al., 2015; Fort et al., 2019) and that the predictive variance amongst component models in these so-called *deep ensembles* is a notion of uncertainty that can be used on downstream decision-making tasks (Lakshminarayanan et al., 2017; Gal et al., 2017; Gustafsson et al., 2020; Ovadia et al., 2019; Yu et al., 2020).

While few theoretical works analyze modern overparameterized ensembles, recent empirical evidence suggests that intuitions from their underparameterized counterparts do not hold in this new regime. For example, classic methods to increase diversity amongst component models, such as bagging, can be harmful for deep ensembles (Nixon et al., 2020; Jeffares et al., 2024; Abe et al., 2022a; 2024; Webb et al., 2021) despite being nearly universally beneficial for underparameterized ensembles. Moreover, while established underparameterized ensembling techniques offer well-founded quantifications of uncertainty (e.g. Mentch & Hooker, 2016; Wager et al., 2014), several recent studies question the reliability of the uncertainty estimates from deep ensembles (Abe et al., 2022b; Theisen et al., 2024; Chen et al., 2024).

To address this divergence and verify recent empirical findings, we develop a theoretical characterization of ensembles in the overparameterized regime, with the goal of contrasting against (traditional) underparameterized ensembles. We answer the following questions:

1. Do ensembles of overparameterized models provide generalization or robustness benefits over a single (very large) model trained on the same data? Does the capacity of the component models affect this difference?

2. What does the predictive variance of overparameterized ensembles measure, and does it relate to classic frequentist or Bayesian notions of uncertainty?

To answer these questions, we analyze ensembles of overparameterized random feature (RF) linear regressors, a theoretically-tractable approximation of neural networks. Unlike prior work on RF models, our analysis makes very few assumptions about the distribution of random features, which—as we will show—is crucial for highlighting the differences between ensemble variance versus more established notions of uncertainty. Our analysis focuses on the practically relevant regime where RF models are trained with little to no regularization. We verify and contextualize our theory with experiments on RF and neural networks ensembles.

## 1.1. Related Work

**Deep ensembles.** A primary motivation of this paper is to understand recent empirical findings about uncertainty quantification afforded by deep ensembles (Lakshminarayanan et al., 2017). Historically, variance amongst deep ensemble members has been a proxy for *epistemic uncertainty* (e.g. Kendall & Gal, 2017; Gustafsson et al., 2020), i.e., the uncertainty that can be reduced by collecting more data. This view reflects a classical intuition of ensembles: ignoring effects of overparameterization and inductive bias, all ensemble members should converge to the same prediction in the infinite data limit, and thus differing predictions suggest a region of the input space with insufficient data. However, recent empirical findings challenge this interpretation of ensemble variance (Abe et al., 2022b; Theisen et al., 2024). Most relevant to our work, Abe et al. (2022b) demonstrate a strong correlation between ensemble variance and the expected improvement that results from increasing model capacity. Specifically, across numerous architectures and datasets, they demonstrate a strong point-wise correlation between the predictions of an ensemble (e.g., 4 ResNet-18s) and a single larger model (e.g., a WideResNet-18 with $4\times$ the width) on both in-distribution and out-of-distribution data. The authors conclude that ensemble variance is more reflective of sensitivity to model capacity rather than data availability, a finding with significant implications for decision-making and robustness. We theoretically verify these findings in ensembles of overparameterized random feature models.

**Random feature models.** The connection between infinitely wide neural networks and kernel methods, particularly Gaussian processes, was pioneered by Neal & Neal (1996) and Williams (1996). Building on these ideas, random feature (RF) models were later introduced as a scalable approximation to kernel machines (Rahimi & Recht, 2007; 2008a;b). RF regressors have seen growing theoretical interest as simplified models of neural networks (e.g. Belkin et al., 2018; 2019; Jacot et al., 2018; Bartlett et al., 2020; Mei & Montanari, 2022; Simon et al., 2024). Random feature models can be interpreted as neural networks where only the last layer is trained (e.g. Rudi & Rosasco, 2017; Belkin et al., 2019) or as first-order Taylor approximations of neural networks (e.g. Jacot et al., 2018).

**Underparameterized random feature models and ensembles.** In this paragraph, we restrict our discussion to analyses of (ensembles of) underparameterized RF regressors, where the number of random features (i.e., the width) is assumed to be far fewer than the number of data points. In the fixed design setting, infinite ensembles of unregularized RF regressors achieve the same generalization error as ridge regression on the original (unprojected) inputs (Kabán, 2014; Thanei et al., 2017; Bach, 2024b). We provide theoretical analysis in Appx. E that further demonstrates ridge-like behaviour of underpameterized RF ensembles.

**Overparameterized random feature models.** Recent works on RF models have focused on the *overparameterized regime*, often using high-dimensional asymptotics to characterize generalization error (Adlam & Pennington, 2020; Bach, 2024b; Hastie et al., 2022; Loureiro et al., 2022; Mei & Montanari, 2022; Ruben et al., 2024). Many works rely on results derived assuming that the the marginal distributions over the random features can be replaced by moment-matched Gaussians. While such approximations are well-founded for asymptotic results (e.g. Goldt et al., 2022; Hu & Lu, 2022; Montanari & Saeed, 2022; Tao, 2012), we argue that they may be harmful specifically for an analysis which aims to characterize the uncertainty properties of ensemble variance. Assuming Gaussianity results in an ensemble variance that is proportional to the predictive variance of Gaussian process regression, often held as a gold standard for uncertainty quantification (Rasmussen & Williams, 2006; Lee et al., 2018; 2020; Ovadia et al., 2019). In contrast, our non-Gaussian analysis yields a characterization of ensemble variance that differs from this conventional notion of uncertainty, closely matching recent empirical studies of ensemble variance (Abe et al., 2022b; Theisen et al., 2024).

The benefits of overparameterization and ensembling for out-of-distribution generalization in random feature models have been analyzed by Hao et al. (2024), who provide lower bounds on OOD risk improvements when increasing capacity or using ensembles. Their work focuses on non-asymptotic guarantees under specific distributional shifts, while ours examines the equivalence of ensembles and sin-

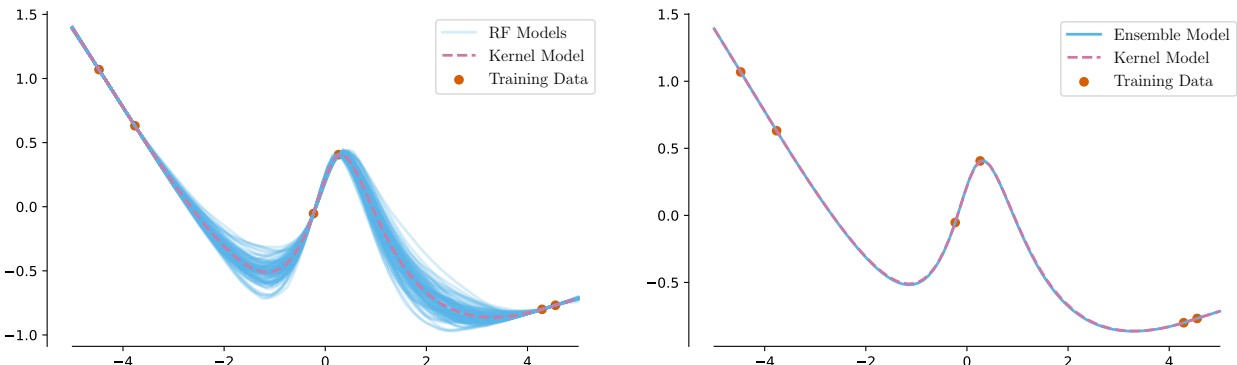

Figure 1: **An infinite ensemble of overparameterized RF models is equivalent to a single infinite-width RF model.** (Left) We show a sample of 100 finite-width RF models (blue) with ReLU activations trained on the same $N = 6$ data points. Additionally, we show the single infinite-width RF model (pink). The finite-width predictions concentrate around the infinite-width model. (Right) We again show the single infinite-width RF model (pink) and the "infinite" ensemble of $M = 10,000$ RF models (blue). We note no perceptible difference between the two in this setting, though extreme numerical conditions can break this equivalence (cf. Fig. 8).

gle large models under minimal assumptions. Concurrent work by Ruben et al. (2024) also finds RF ensembles offer little advantage over larger single models, though their analysis uses optimal ridge tuning and Gaussian universality assumptions. Most related to our work is Jacot et al. (2020), who analyze the pointwise expectation and variance of ridge-regularized RF models with Gaussian process (GP) features, leveraging Gaussianity to simplify their analysis. We go beyond this prior work by significantly weakening the assumptions on the distribution of random features, enabling us to characterize differences between ensembles versus Gaussian models with respect to uncertainty and robustness properties. Moreover, we provide a finite-sample analysis as well as a characterization of the transition from the ridgeless to ridge-regularized regimes, which—to the best of our knowledge—are novel results for overparameterized RF ensembles.

## 1.2. Contributions

We consider ensembles of *overparameterized* RF regressors in both the ridgeless and small ridge regimes. Unlike prior work, we make minimal assumptions about the distribution of the random features and so our results are not restricted to high-dimensional asymptotics where Gaussian universality might typically apply. Our results thus distinguish differences between RF ensembles and more traditional uncertainty-aware models like Gaussian processes. Concretely, we make the following contributions:

To answer Question 1: we show that the average ridgeless RF regressor is pointwise equivalent to its corresponding ridgeless kernel regressor (Theorem 3.2), implying that an infinite ensemble of overparameterized RF models is *exactly*

equivalent to a single infinite-width RF model (cf. Fig. 1). We further show that this equivalence approximately holds in the small ridge regime (Theorem 3.5). Moreover, we extend these results to a finite parameter budget, showing that the functional difference between the parameters of a larger single model and a finite ensemble, each with the same total number of parameters, is small with high probability (see Sec. 3.2). We validate these theoretical results with supporting experiments on RF and neural network ensembles, using synthetic data and the California Housing dataset (Kelley Pace & Barry, 1997) with various activation functions (detailed in Appx. A.1 and Appx. B).

To answer Question 2: we show that the predictive variance in an overparameterized ensemble generally does not have a frequentist or Bayesian interpretation, unlike uncertainty quantifications obtained from Gaussian processes. Instead, we find that the variance measures the expected squared difference between the predictions from a (finite-width) RF regressor and its corresponding kernel regressor (i.e., the infinite-width model) (see Sec. 3.3). Crucially, this finding relies on our non-Gaussian analysis of RF models.

Altogether, these results support recent empirical findings that deep ensembles offer few generalization and uncertainty quantification benefits over larger single models (Abe et al., 2022b; Theisen et al., 2024). Our theory and experiments demonstrate that these phenomena are not specific to neural networks or Gaussian models but are more general properties of ensembles in the overparameterized regime.

## 2. Setup

We work in a regression setting. The training dataset $\mathcal{D} = \{(x_i, y_i)\}_{i=1}^N \in (\mathcal{X} \times \mathbb{R})^N$ is a *fixed* set of size $N$. The vector $y \in \mathbb{R}^N$ is the concatenation of all training responses.

We consider *RF models* adhering to the form $h_{\mathcal{W}}(x) = \frac{1}{\sqrt{D}} \sum_{i=1}^D \phi(\omega_i, x) \theta_i$, where $\theta_i$ are learned parameters, $\mathcal{W} = \{\omega_i\}_{i=1}^D \in \Omega^D$ are i.i.d. draws from some distribution $\pi(\cdot)$, and $\phi : \Omega \times \mathcal{X} \to \mathbb{R}$ is a *feature extraction function*. In the case of a ReLU-based RF model with $p$-dimensional inputs, we have $\mathcal{X} = \Omega = \mathbb{R}^p$ and $\phi(\omega_i, x) = \max(0, \omega_i^\top x)$. Though RF models cannot fully explain the behaviour of neural networks (e.g. Ghorbani et al., 2019; Li et al., 2021; Pleiss & Cunningham, 2021), they can be a useful proxy for understanding the effects of overparameterization and capacity on generalization (e.g. Belkin et al., 2019; Adlam & Pennington, 2020; Mallinar et al., 2022).

**Notation.** For any $x, x' \in \mathcal{X}$, we denote the second moment of the feature extraction function $\phi(\omega, \cdot)$ as $k(x, x') = \mathbb{E}_\omega[\phi(\omega, x)\phi(\omega, x')]$, which is a positive definite kernel function. We use the matrix $K := [k(x_i, x_j)]_{ij} \in \mathbb{R}^{N \times N}$ for the kernel function applied to all training data pairs and the matrix $\Phi_{\mathcal{W}} := [\phi(\omega_j, x_i)]_{ij} \in \mathbb{R}^{N \times D}$ for the feature extraction function applied to all data/feature combinations. In this notation, $[\cdot]_{ij}$ refers to the entry in the $i$-th row and $j$-th column; if one index is omitted (e.g., $[v]_j$), it refers to the $j$-th element of a row- or column-vector, depending on the context. We drop the subscript $\mathcal{W}$ when the set of random features is clear from context. Furthermore, we assume that $K$ is invertible.

Throughout our analysis, it will be useful to consider the "whitened" features $W = R^{-\top} \Phi \in \mathbb{R}^{N \times D}$ where $R^\top R = K$ is the Cholesky decomposition of the kernel matrix $K$. When considering a test point $x^* \in \mathcal{X}$ (or equivalently a set of test points), we extend the $K, R, \Phi, W$ notation by

$$\begin{bmatrix} K & [k(x_i, x^*)]_i \\ [k(x^*, x_j)]_j & k(x^*, x^*) \end{bmatrix} = \begin{bmatrix} R & c \\ 0 & r_\perp \end{bmatrix}^\top \begin{bmatrix} R & c \\ 0 & r_\perp \end{bmatrix},$$

$$\begin{bmatrix} W \\ w_\perp^\top \end{bmatrix} = \begin{bmatrix} R & c \\ 0 & r_\perp \end{bmatrix}^{-\top} \begin{bmatrix} \Phi \\ [\phi(\omega_i, x^*)]_i \end{bmatrix}. \quad (1)$$

For fixed training/test points, $\mathbb{E}_W[WW^\top] = D \cdot I$, $\mathbb{E}_{w_\perp}[w_\perp^\top w_\perp] = D$ and $\mathbb{E}_{W,w_\perp}[w_\perp^\top W^\top] = 0$ which can be directly derived from $\mathbb{E}_\Phi[\Phi\Phi^\top] = D \cdot K$ (and similar properties for $\phi^*$, the vector of feature evaluations at $x^*$, i.e., $[\phi(\omega_j, x^*)]_j$). Moreover, the columns $[w_i; w_{\perp i}]$ of $[W; w_\perp]$ are i.i.d. since they are affine transformations of the i.i.d. columns of $\Phi$.

**Overparameterized ridge/ridgeless regressors and ensembles.** As our focus is the overparameterized regime, we assume a computational budget of $D > N$ features

$(\mathcal{W} = \{\omega_1, \ldots, \omega_D\} \sim \pi^D)$ to construct an RF regressor $h_{\mathcal{W}}(x) = \frac{1}{\sqrt{D}} \phi_{\mathcal{W}}(x)^\top \theta$. We train the regressor parameters $\theta$ to minimize the loss $\|\frac{1}{\sqrt{D}} \Phi_{\mathcal{W}} \theta - y\|_2^2 + \lambda \|\theta\|_2^2$ for some ridge parameter $\lambda \geq 0$. When $\lambda > 0$ this optimization problem admits the closed-form solution $\theta_{\mathcal{W}, \lambda}^{(\mathrm{RR})} = \frac{1}{\sqrt{D}} \Phi_{\mathcal{W}}^\top \left(\frac{1}{D} \cdot \Phi_{\mathcal{W}} \Phi_{\mathcal{W}}^\top + \lambda I\right)^{-1} y$. Although the learning problem is underspecified when $\lambda = 0$ (i.e. in the ridgeless case), the implicit bias of (stochastic) gradient descent initialized at zero leads to the minimum norm interpolating solution $\theta_{\mathcal{W}}^{(\mathrm{LN})} = \frac{1}{\sqrt{D}} (\Phi)^\top \left(\frac{1}{D} \cdot \Phi\Phi^\top\right)^{-1} y$. We denote the resulting ridge(less) regressors as $h_{\mathcal{W}}^{(\mathrm{LN})}(\cdot) := \frac{1}{\sqrt{D}} [\phi(\omega_j, \cdot)]_j \theta_{\mathcal{W}}^{(\mathrm{LN})}$, and $h_{\mathcal{W}, \lambda}^{(\mathrm{RR})}(\cdot) := \frac{1}{\sqrt{D}} [\phi(\omega_j, \cdot)]_j \theta_{\mathcal{W}, \lambda}^{(\mathrm{RR})}$.

We also consider ensembles of $M$ ridge(less) regressors. We assume that each is trained on a different set of i.i.d. $D > N$ random features $\mathcal{W}_1, \ldots, \mathcal{W}_M \sim \pi^D$ but trained on the same training set. Thus, the only source of randomness in these ensembles comes from the random selection of features $\mathcal{W}_i$, analogous to the standard training procedure of deep ensembles (Lakshminarayanan et al., 2017). The ensemble prediction is given by the arithmetic average of the individual models $\bar{h}_{\mathcal{W}_{1:M}}(\cdot) = \frac{1}{M} \sum_{m=1}^M h_{\mathcal{W}_m}^{(\mathrm{LN})}(\cdot)$.

**Assumptions.** A key difference between this paper and prior literature is the set of assumptions about the random feature distribution $\pi(\cdot)$. Most prior works assume that entries in the extended whitened feature matrix $[W; w_\perp]$ are i.i.d. draws from standard normal distribution (e.g. Adlam & Pennington, 2020; Jacot et al., 2020; Mei & Montanari, 2022; Simon et al., 2024) implying that $\phi(\omega_i, \cdot)$ are draws from a Gaussian process with covariance $k$.[1] While Gaussianity is appropriate in high-dimensional asymptotics, it essentially reduces analysis about the ensemble distribution to a statement about Gaussian processes. A major focus of this work is to differentiate ensembles from Gaussian processes with regards to uncertainty quantification.

Even if we were to relax the Gaussian assumption to a sub-Gaussian assumption, (as done by Bartlett et al., 2020; Bach, 2024a), the distribution of random features will still not accurately reflect common neural network features if the entries of $[W; w_\perp]$ are assumed to be i.i.d. For instance, consider ReLU features. If $\mathcal{X} \subseteq \mathbb{R}^p$ with $p < N$, the function $\max(\omega^\top x, 0)$ can be fully specified by a $p$-dimensional random variable. Thus, knowing $N$ evaluations of $\omega_j^\top x_i$ allows one to infer $\omega_j$, making $w_\perp$ deterministic given $W$. We instead consider the following less restrictive assumptions on the distribution of random feature functions $\pi(\cdot)$:

---

[1] If the entries of $W, w_\perp$ are i.i.d. Gaussian, then the $i^{\mathrm{th}}$ feature applied to train/test inputs ($[R^\top w_i; c^\top w_i + r_\perp w_{\perp i}]$) is multivariate Gaussian. This fact holds for any train/test data; thus the $i^{\mathrm{th}}$ feature is a GP by definition (e.g. Rasmussen & Williams, 2006, Ch. 2).

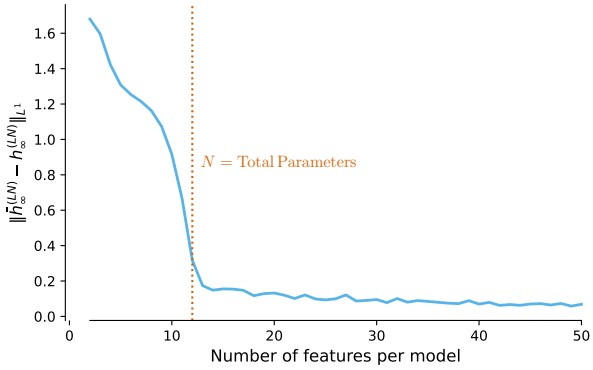 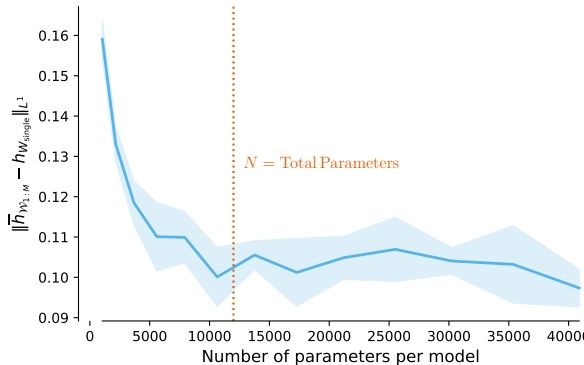

Figure 2: **Overparameterized ensembles are equivalent to a single infinite-width model regardless of feature distribution, while underparameterized ensembles behave differently.** We present the average absolute difference between large ensembles of models with $D$ features versus a single large (or infinite) width model. **(Left) RF ensembles** with softplus activations, $N = 12$, using the California Housing dataset (Kelley Pace & Barry, 1997). **(Right) Neural network ensembles** with ReLU activations, $N = 12,000$, on the same dataset. The shaded region shows the standard deviation. Both exhibit a "hockey stick"-like pattern, less pronounced for neural networks, where the difference between underparameterized ensembles and the large model is substantial, but diminishes for $D > N$.

**Assumption 2.1** (Assumption of subexponentiality).

1. $w_i w_{\perp i}$ (where $w_i$ is the $i^{\text{th}}$ column of $W$) is subexponential $\forall i \in \{1, ..., D\}$ and

2. $\sum_{i=1}^{D} w_i w_i^\top$ is a.s. positive definite for any $D \geq N$.

The first condition of Assumption 2.1 ensures that the whitened random features do not have excessively "heavy tails", meaning their values are well-concentrated. This is a mild condition, satisfied if the individual feature components $w_i$ and $w_{\perp i}$ are sub-Gaussian (but potentially dependent), which is true if the features come from activation functions with bounded derivatives and sub-Gaussian weights. The second condition is equivalent to $\Phi$ having almost surely full rank, which is not true for ReLUs and leaky-ReLUs features but which is true for arbitrarily precise approximations thereof.[2] Note we make no assumptions about the mean or independence of the entries in a given column of $[W; w_\perp]$.

## 3. Main results

### 3.1. Equivalence of Infinite Ensembles and the Infinite-Width Single Models

We at first assume an infinite computational budget and consider the following two limiting predictors, for which we will show pointwise equivalence in predictions:

1. An infinite-width least norm predictor, $h_\infty^{(\text{LN})}$, the a.s.

---

[2]E.g., $\phi_\alpha(\omega, x) = \frac{1}{\alpha} \log(1 + e^{\alpha \omega^\top x})$, $\alpha > 0$ yields an a.s. full-rank $\Phi$, and $\phi_\alpha(\omega, x) \overset{\alpha \to \infty}{\to} \text{ReLU}(\omega^\top x)$.

limit of $h_{\mathcal{W}}^{(\text{LN})}$ as $|\mathcal{W}| = D \to \infty$

2. An infinite ensemble of finite-width least norm predictors, $\bar{h}_\infty^{(LN)}$, which is the almost sure limit of $\bar{h}_{\mathcal{W}_{1:M}}^{(LN)}$ as $M \to \infty$, with $N < D < \infty$ remaining constant.

These limiting predictors not only approximate large ensembles and very large single models but also help characterize the variance and generalization error of finite overparameterized ensembles, as discussed in Sec. 3.3.

Define $k_N(\cdot) : \mathcal{X} \to \mathbb{R}^N$ as the vector of kernel evaluations with the training data $k_N(\cdot) = \begin{bmatrix} k(x_1, \cdot) & \cdots & k(x_N, \cdot) \end{bmatrix}^\top$. As $D \to \infty$, the minimum norm interpolating model converges pointwise almost surely to the ridgeless kernel regressor by the Strong Law of Large Numbers:

$$h_{\mathcal{W}}^{(\text{LN})}(\cdot) \overset{\text{a.s.}}{\longrightarrow} h_\infty^{(\text{LN})}(\cdot), \qquad h_\infty^{(\text{LN})}(\cdot) := k_N(\cdot)^\top K^{-1} y.$$

On the other hand, using $W$ and $w_\perp$ as introduced in Sec. 2 we can rewrite the infinite ensemble prediction $\bar{h}_\infty^{(LN)}(x^*)$ as (for a derivation of this, see Appx. C.1)

$$\bar{h}_\infty^{(LN)}(x^*) = h_\infty^{(\text{LN})}(x^*)$$
$$+ \ r_\perp \mathbb{E}_{W, w_\perp} \left[ w_\perp^\top W^\top \left( W W^\top \right)^{-1} \right] R^{-\top} y \quad (2)$$

To prove the pointwise equivalence of the infinite ensemble and infinite-width single model, we need to show that $\mathbb{E}_{W, w_\perp}[w_\perp^\top W^\top (W W^\top)^{-1}]$ term in Eq. (2) is zero. Note that this result trivially holds when the entries of $W$ and $w_\perp$ are i.i.d. and zero mean, as assumed in prior work (e.g. Jacot et al., 2020). In the following lemma, we show that this

term is zero even when $w_\perp$ and $W$ are dependent, which—as described in Sec. 2—is a more realistic assumption for neural network features:

**Lemma 3.1.** *Under Assumption 2.1, it holds that* $\mathbb{E}_{W,w_\perp}[w_\perp^\top W^\top (WW^\top)^{-1}] = 0$.

(Proof: see Appx. C.1.) Combining Lemma 3.1 and Eq. (2) yields the pointwise equivalence of $\bar{h}_\infty^{(LN)}$ and $h_\infty^{(LN)}$:

**Theorem 3.2** (Equivalence of infinite-width single model and infinite ensembles)**.** *Under Assumption 2.1, the infinite ensemble of finite-width (but overparameterized) RF regressors $\bar{h}_\infty^{(LN)}$ is pointwise almost surely equivalent to the (single) infinite-width RF regressor $h_\infty^{(LN)}$.*

Theorem 3.2 implies that ensembling overparameterized RF models yields *exactly* the same predictions as simply increasing the capacity of a single RF model, regardless of the RF distribution (see Fig. 1 for a visualization). Note that this result significantly generalizes prior characterizations of overparameterized RF models that have relied on Gaussianity assumptions or asymptotic analyses (e.g. Adlam & Pennington, 2020; Jacot et al., 2020), demonstrating that the ensemble/infinite-single model equivalence is a fundamental property of overparameterization. Consequently, we should not expect substantial differences in generalization between large single models and overparameterized ensembles, consistent with recent empirical findings by (Abe et al., 2022b; 2024; Theisen et al., 2024).

We emphasize a contrast with the underparameterized regime, where RF ensembles match the generalization error of kernel ridge regression (see Appx. E or Bach, 2024b, Sec. 10.2.2). Width controls the implicit ridge parameter in the underparameterized regime (see Sec. 1.1), whereas width does not affect the ensemble predictor in the overparameterized regime. We confirm this difference in Fig. 2 which shows that RF ensembles are close to the ridgeless kernel regressor when $D > N$ but not when $D < N$. The figure also illustrates similar behaviours when comparing deep ensembles versus single large neural networks.

## 3.2. Ensembles versus Larger Single Models under a Finite Parameter Budget

For modest parameter budgets (where our asymptotic results are not applicable), we compare whether ensembles are more parameter efficient than larger single models. Specifically, given access to $MD$ random features, we compare ensembles of $M$ models each of which use $D$ of the features $(\bar{h}_{\mathcal{W}_{1:M}}^{(LN)} = \frac{1}{M}\sum_{m=1}^M h_{\mathcal{W}_m}^{(LN)})$ against a single model that uses all $MD$ features $h_{\mathcal{W}^*}^{(LN)}(\cdot)$ (i.e., here $|\mathcal{W}_m| = D$ for all $m$ and $|\mathcal{W}^*| = MD$).

First, we provide a non-asymptotic theorem showing that $\bar{h}_{\mathcal{W}_{1:M}}^{(LN)}$ and $h_{\mathcal{W}^*}^{(LN)}$ behave similarly, with their difference

becoming negligible as the number of features per ensemble member increases (for a formal version, see Appx. C.2).

**Theorem 3.3** (Non-asymptotic difference between ensembles and single models (informal version))**.** *Under slightly stronger assumptions than Assumption 2.1, the $L_2$ difference between a single neural network with $MD$ features and an ensemble of $M$ neural networks each with $D$ features is, with probability $1 - \delta$, upper bounded by:*

$$\left\| h_{\mathcal{W}^*}^{(LN)}(\cdot) - \bar{h}_{\mathcal{W}_{1:M}}^{(LN)}(\cdot) \right\|_2^2 \leq O(\sqrt{\log(1/\delta)}) + O(1/D)$$

Theorem 3.3 is supported through a standard bias-variance decomposition of risk:

$$\mathbb{E}_h\left[L\left(h\right)\right] := \mathbb{E}_h\left[\mathbb{E}_x[(h(x) - \mathbb{E}[y \mid x])^2]\right.$$
$$= L\left(\mathbb{E}_h\left[h\right]\right) + \mathbb{E}_x\left[\mathbb{V}_h\left(h(x)\right)\right]. \quad (3)$$

Since $h_{\mathcal{W}^*}^{(LN)}$ and $h_{\mathcal{W}_1}^{(LN)}, \ldots, h_{\mathcal{W}_M}^{(LN)}$ share the same expected predictor (as established in Theorem 3.2), the only difference in the generalization of $h_{\mathcal{W}^*}^{(LN)}$ and $\bar{h}_{\mathcal{W}_{1:M}}^{(LN)}$ arises from their variances. Due to the independence between ensemble members, we have that $\mathbb{V}_{\mathcal{W}_{1:M}}[\bar{h}_{\mathcal{W}_{1:M}}^{(LN)}(x)] = \frac{1}{M}\mathbb{V}_{\mathcal{W}_m}[h_{\mathcal{W}_m}^{(LN)}(x)]$. Moreover, prior works such as (Adlam & Pennington, 2020) and empirical results (see Appx. A.3) suggest the variance of a single RF model is inversely proportional to the number of features.[3] As a consequence, we have that $\mathbb{V}_{\mathcal{W}^*}[h_{\mathcal{W}^*}^{(LN)}(x)] : \mathbb{V}_{\mathcal{W}_m}[h_{\mathcal{W}_m}^{(LN)}(x)] \asymp 1/M$, further suggesting that the generalization of ensembles and single models should be similar under the same parameter budget.

Fig. 3 (left) and Appx. A.3 empirically confirm that RF ensembles versus single RF models obtain similar generalization under fixed feature budgets. Moreover, Fig. 3 (right) depicts a similar trend for neural networks: deep ensembles perform roughly the same as larger single models under a fixed parameter budget. These results show that ensembles offer no meaningful generalization advantage over (large) single models and, since the arguments hold for any test distribution, align with empirical findings (Abe et al., 2022b) that ensembles provide no additional robustness benefits.

## 3.3. Implications for Uncertainty Quantification

We now analyze the predictive variance amongst component models in an overparameterized RF ensemble, a quantity often used to quantify predictive uncertainty in safety-critical applications (Lakshminarayanan et al., 2017). Before diving in to a mathematical characterization, it is worth reflecting on the qualitative characterization based on our existing results. Because the expected overparameterized RF model

---

[3]This rate is exact for Gaussian features and approximate for the general case.

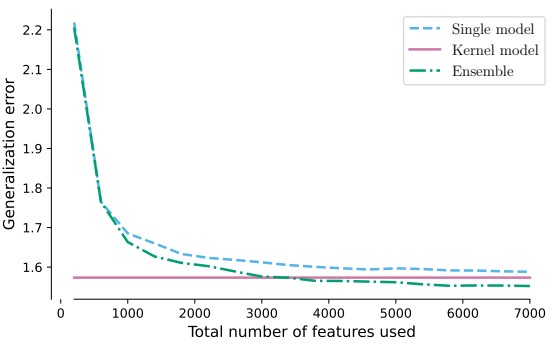 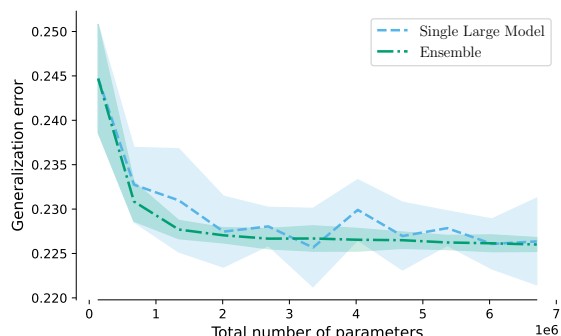

Figure 3: **The generalization error of overparameterized ensembles and single large models scales similarly with the total number of features.** We present the generalization error of ensembles compared to single large models of the same type with equivalent total parameter budgets. Both exhibit nearly identical dependence on the total feature budget. **Left: RF ensembles/models**, $N = 12$, ReLU activations, ensembles of models with $D = 200$ features. **Right: neural networks ensembles/models**, $N = 12,000$, ensembles of three-layer MLPs with width 256 in each layer. The shaded region shows the standard deviation.

is the infinite-width model, predictive variance is equal to

$$\mathbb{V}_{\mathcal{W}}[h_{\mathcal{W}}^{(\text{LN})}(x^*)] = \mathbb{E}_{\mathcal{W}}\left[\left(h_{\mathcal{W}}^{(\text{LN})}(x^*) - \overbrace{\bar{h}_{\infty}^{(\text{LN})}(x^*)}^{\mathbb{E}_{\mathcal{W}}[h_{\mathcal{W}}^{(\text{LN})}(x^*)]}\right)^2\right]$$
$$= \mathbb{E}_{\mathcal{W}}\left[\left(h_{\mathcal{W}}^{(\text{LN})}(x^*) - h_{\infty}^{(\text{LN})}(x^*)\right)^2\right],$$

i.e. the expected difference between finite- versus infinite-width RF model predictions. In other words, *ensemble variance quantifies how predictions change if we increase model capacity.* This characterization, which holds for all random feature distributions satisfying Assumption 2.1, is not a standard frequentist or Bayesian notion of uncertainty except under specific distributional assumptions.

**Uncertainty quantification under Gaussian features.** Using Theorem 3.2, the variance of the predictions of a single RF model with respect to its random features can be expressed as (see Appx. C.3 for a derivation)

$$\mathbb{V}_{\mathcal{W}}[h_{\mathcal{W}}^{(\text{LN})}(x^*)] = r_{\perp}^2 \left(y^{\top}R^{-1}\,\mathbb{E}_{W,w_{\perp}}\left[(WW^{\top})^{-T}Ww_{\perp}\right.\right.$$
$$\left.\left. \cdot\, w_{\perp}^{\top}W^{\top}(WW^{\top})^{-1}\right]R^{-\top}y\right). \qquad (4)$$

In the special case where $W$ and $w_{\perp}$ are i.i.d. standard normal, this expression simplifies to

$$\mathbb{V}_{\mathcal{W}}[h_{\mathcal{W}}^{(\text{LN})}(x^*)] = r_{\perp}^2 \left(\frac{\|h_{\infty}^{(\text{LN})}\|_k^2}{D-N-1}\right), \qquad (5)$$

where $\|h_{\infty}^{(\text{LN})}\|_K^2$ represents the squared norm of $h_{\infty}^{(\text{LN})}$ in the RKHS defined by the kernel $k(\cdot,\cdot)$. From this equation, we see that $\mathbb{V}_{\mathcal{W}}[h_{\mathcal{W}}^{(\text{LN})}(x^*)]$ only depends on $x^*$ through the quantity $r_{\perp}^2$, which by Eq. (1) is equal to

$$r_{\perp}^2 = k(x^*, x^*) - k_N(x^*)^{\top}K^{-1}k_N(x^*).$$

We recognize this quantity as the Gaussian process posterior variance with prior covariance $k(\cdot,\cdot)$ (e.g. Rasmussen & Williams, 2006). Thus, with Gaussian features, ensemble variance admits a Bayesian interpretation in addition to the model capacity interpretation. In other words, Gaussianity assumptions justify the use of overparameterized ensemble variance in uncertainty quantification tasks.

**Uncertainty quantification under general features.** Unfortunately, this Bayesian interpretation explicitly does not carry over to the general Assumption 2.1 case. Although Eq. (4) still holds for general feature distributions, it does not have a simple expression unless $W$ and $w_{\perp}$ are independent. The variance depends on $x^*$ through both $r_{\perp}^2$ as well as through a complicated expectation involving $W$ and $w_{\perp}$. In Appx. C.3 we demonstrate with a simple example that this expectation can indeed depend on $x^*$, implying that ensemble variance does not correspond to a scalar multiple of $r_{\perp}^2$ (i.e. the Gaussian process posterior variance).

In numerical experiments using ReLU random features, (Fig. 4 and Appx. A.3), we observe significant deviations between the ensemble variance and the Gaussian process posterior variance, further suggesting that one cannot view ensembles through a classic framework of uncertainty. These discrepancies are particularly important for uncertainty estimation in safety-critical applications or active learning (e.g. Gal et al., 2017; Beluch et al., 2018), as the only meaningful interpretation of ensemble variance (the expected change in prediction from increasing capacity) may not yield reliability guarantees or useful exploration-exploitation tradeoffs.

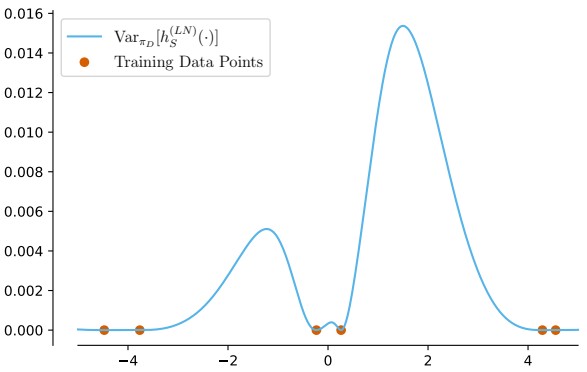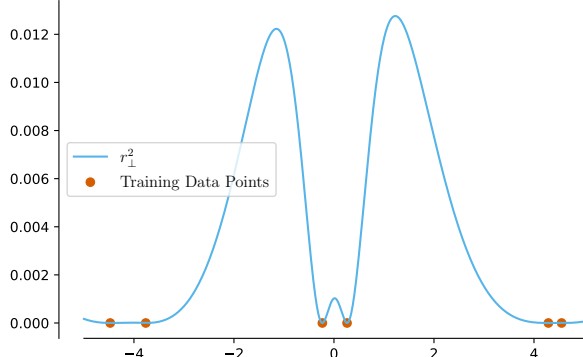

Figure 4: **RF ensemble variance (left) and Bayesian notions of uncertainty (right) can differ significantly.** For $N = 6$ and $D = 200$ with ReLU activations, the overparameterized ensemble variance (left) and the posterior variance of a Gaussian process with prior covariance $k(\cdot, \cdot)$ (right) differ substantially across the input range.

### 3.4. Equivalence of the Limiting Predictors in the Small Ridge Regime

Having established the equivalence between infinite ensembles and infinite-width single models in the ridgeless regime, we now investigate whether this equivalence approximately persists in the practically relevant setting when a small ridge regularization parameter $\lambda > 0$ is introduced. More generally, we aim to determine whether the transition from the ridgeless case to the small ridge regime is smooth. While $h_{\infty,\lambda}^{(RR)}$, the infinite-width limit of $h_{\mathcal{W},\lambda}^{(RR)}$ as $|\mathcal{W}| = D \to \infty$, almost surely converges to the kernel ridge regressor with ridge $\lambda$, the infinite ensemble $\bar{h}_{\infty,\lambda}^{(RR)} := \mathbb{E}_{\mathcal{W}}[h_{\mathcal{W},\lambda}^{(RR)}(x)]$ does not generally maintain pointwise equivalence with $h_{\infty,\lambda}^{(RR)}$. This divergence occurs even under Gaussianity assumptions (Jacot et al., 2020). However, we hypothesize that the difference between these limiting predictors is small when $\lambda$ is close to zero, which is common in practical applications. To analyze this regime, we introduce a minor additional assumption, which is weaker than Gaussianity:

**Assumption 3.4.** We assume that $\mathbb{E}_{\mathcal{W}}[(\Phi_{\mathcal{W}}\Phi_{\mathcal{W}}^{\top})^{-1}]$ is finite for all $|\mathcal{W}| = D > N$.

Under Assumptions 2.1 and 3.4, we show that the difference between ridge-regularized ensembles and single models is Lipschitz-continuous with respect to $\lambda$ (proof in Appx. D.1).

**Theorem 3.5** (The difference between ensembles and large single models is smooth with respect to $\lambda$.). *Under Assumptions 2.1 and 3.4, the difference $|\bar{h}_{\infty,\lambda}^{(RR)}(x^*) - h_{\infty,\lambda}^{(RR)}(x^*)|$ between the infinite ensemble and the single infinite-width model trained with ridge $\lambda \geq 0$ is Lipschitz-continuous in $\lambda$. The Lipschitz constant is independent of $x^*$ for compact $\mathcal{X}$.*

This result is illustrated in Fig. 5, where the terms bounding the difference evolve smoothly with $\lambda$. To the best of

our knowledge, this Lipschitz-continuity has not been established even under Gaussianity assumptions. We note that the bound by Jacot et al. (2020, Thm. 4.1), which characterizes the difference between ridge ensembles and infinite models with Gaussian random features, becomes vacuous as $\lambda \to 0$. Since Theorem 3.2 ensures that $|\bar{h}_{\infty,\lambda}^{(RR)}(x^*) - h_{\infty,\lambda}^{(RR)}(x^*)| = 0$ for $\lambda = 0$, we can conclude that the pointwise difference grows at most linearly with $\lambda$. Specifically, we have that

$$\left| \bar{h}_{\infty,\lambda}^{(RR)} - h_{\infty,\lambda}^{(RR)}(x) \right| \leq C \cdot \lambda,$$

for some constant $C$ independent of $x^*$, provided that $\mathcal{X}$ is compact. In practical terms, this result indicates that for sufficiently small values of $\lambda$, the predictions of large ensembles and large single models remain nearly indistinguishable, reinforcing our findings from the ridgeless regime.

## 4. Conclusion

This work characterized overparameterized RF ensembles and contextualized theoretical findings with neural network experiments. We used weaker distributional assumptions than prior work to (a) more faithfully approximate real-world models and (b) highlight differences between ensembles versus models with Gaussian behaviour.

For *Question 1*, we demonstrated under weak conditions that infinite ensembles and single infinite-width models are pointwise equivalent in the ridgeless regime (Theorem 3.2) and nearly identical with a small ridge (Theorem 3.5), significantly expanding on prior results. We further provide a non-asymptotic characterization, showing that ensembles and large single models with the same parameter budget are nearly equivalent (Theorem 3.3). These results verify recent empirical findings (e.g. Abe et al., 2022b) that much

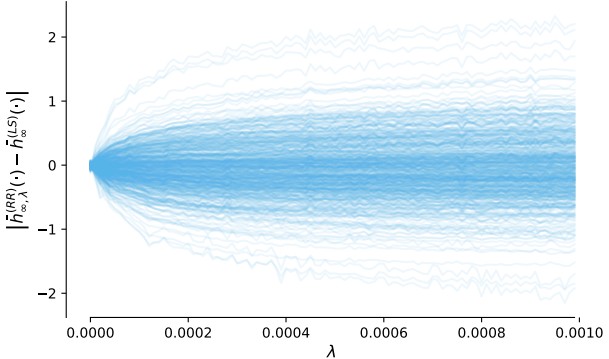 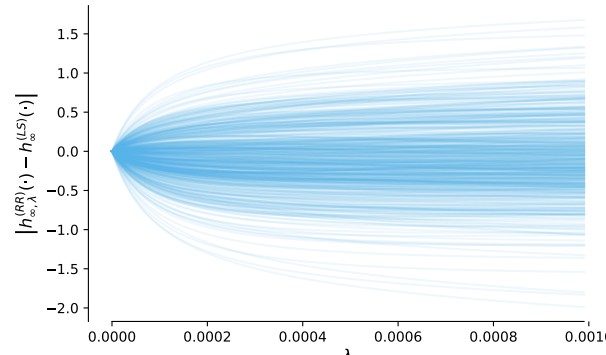

Figure 5: **Lipschitz continuity of predictions for an infinite ensemble and kernel regressor with respect to the ridge parameter.** (Left) We plot $|\bar{h}_{\infty,\lambda}^{(RR)}(x^*) - \bar{h}_{\infty}^{(LS)}(x^*)|$ as a function of $\lambda$ for 500 test points. (Right) We show the evolution of $|h_{\infty,\lambda}^{(RR)}(x^*) - h_{\infty}^{(LS)}(x^*)|$ for the same test points. Both plots use ReLU activation functions and the California Housing Dataset with $N = 12$ and $D = 200$. While the direct difference $|\bar{h}_{\infty,\lambda}^{(RR)}(x^*) - h_{\infty,\lambda}^{(RR)}(x^*)|$ is not shown (for reasons outlined in Appx. A.4), it is bounded by the sum of the plotted quantities (see Appx. D.1). The evolution of these plotted bounding terms thus illustrates that this direct difference is Lipschitz continuous in $\lambda$ (proven in Theorem 3.5) and converges to zero as $\lambda \to 0$ (a consequence of Theorem 3.5 and the ridgeless equivalence established in Theorem 3.2).

of the benefit attributed to overparameterized ensembles, such as improved predictive performance and robustness, can be explained by their similarity to larger single models. Notably, our analysis does not rely on Gaussianity, emphasizing that these phenomena are fundamental properties of overparameterized models and not artifacts of specific feature assumptions.

In contrast, for *Question 2*, we found that uncertainty interpretations of ensemble variance are contingent on Gaussianity assumptions and fall apart under more general feature distributions. We characterize ensemble variance as the expected difference to a single larger model, which only corresponds to a (scaled) Bayesian notion of uncertainty under strong independence assumptions. With more realistic feature distributions, the ensemble variance does not correspond to any conventional notion of uncertainty, reinforcing recent empirical findings on the limitations of ensemble uncertainty quantification (Abe et al., 2022b). This deviation supplies further evidence that caution is needed when using ensembles in safety-critical settings.

Overall, while our results do not contradict the utility of overparameterized ensembles, they suggest that their benefits may often be explained by their similarity to larger models and that further research is needed to improve uncertainty quantification methods.

## Impact Statement

This paper presents work whose goal is to advance the field of Machine Learning. There are many potential societal

consequences of our work, none which we feel must be specifically highlighted here.

## Acknowledgements

GP is supported by Canada CIFAR AI Chairs. JPC is supported by the Gatsby Charitable Foundation (GAT3708), the Simons Foundation (542963), the NSF AI Institute for Artificial and Natural Intelligence (ARNI: NSF DBI 2229929) and the Kavli Foundation. We acknowledge the support of the Natural Sciences and Engineering Research Council of Canada (NSERC: RGPIN-2024-06405). Resources used in preparing this research were provided, in part, by the Province of Ontario, the Government of Canada through CIFAR, and companies sponsoring the Vector Institute.

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

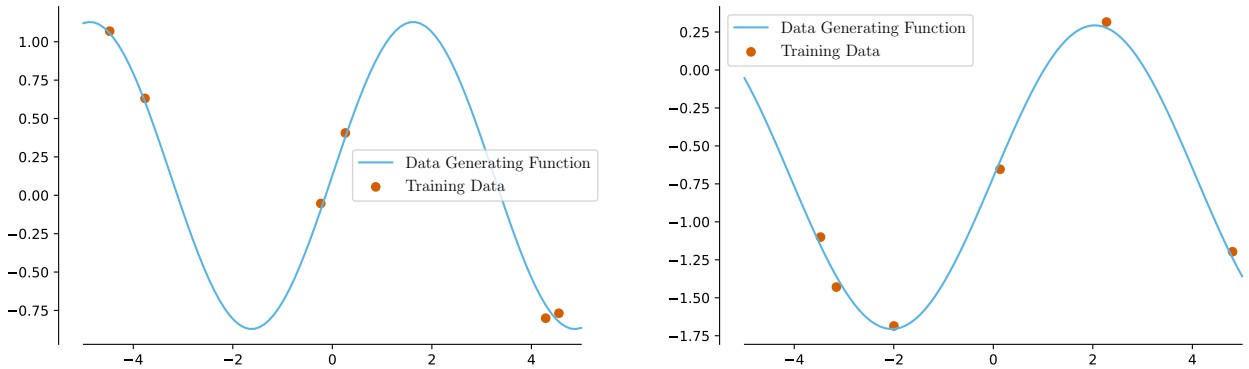

Figure 6: **True function** $f(x) = \sin(5 \cdot b^\top x)$ **with different random seeds.** The blue line shows the true function, while red dots represent training samples for two distinct random seeds.

In the appendix, we will provide the following additional results:

1. In Appx. A, we will describe our experimental setup for RF models in more detail, difficulties we encountered when developing the experiments, and provide the results of additional experiments.

2. In Appx. B, we will describe our experimental setup for neural network models in more detail, and provide the results of additional experiments.

3. In Appx. C we will give the proofs for Secs. 3.1 and 3.3 in the main paper.

4. In Appx. D we will give the proofs for Sec. 3.4 in the main paper.

5. Finally, in Appx. E, we prove (under mild assumptions) that infinite underparameterized RF ensembles are equivalent to kernel ridge regression under some transformed kernel.

The code to run all our experiments can be found on GitHub: https://github.com/nic-dern/theoretical-limitations-overparameterized-ensembles. It contains a README.md file that explains how to set up and run the experiments.

## A. Experimental Setup and Additional Results for RF Models

### A.1. Experimental Setup

We had two setups using which we performed most of our experiments:

1. We generate training and test points uniformly at random from $[-5, 5]^d$ using the function $f(x) = \sin(5 \cdot b^\top x)$, where $b$ is a vector (depending on the random seed) and the noise parameter is $\sigma = 0.05$ (we assume Gaussian noise with mean 0). In this setting, we use $N = 6, D = 200$, and data from $\mathbb{R}$ (i.e., $d = 1$) if not specified otherwise. You can find a plot of an example true function in Fig. 6.

2. We use the California Housing (Kelley Pace & Barry, 1997) dataset and sample distinct training and test points from it (randomly permuting the dataset initially). In this setting, we use $N = 12, D = 200$ if not differently specified. The data dimension is $\mathbb{R}^8$ here. In contrast to the first setting, we employ a data normalization using a max-min normalization *on the entire dataset* since we experimentally found this makes our methods more stable.

We calculate the generalization error using $N = 1000$ test points in both settings. In the first setting, we calculate the variance of the predictions of a single model using $M = 20,000$ models, while in the second setting, we use $M = 4,000$ models. Apart from Fig. 12 where we use $100,000$ samples, "infinite" ensembles consist of $M = 10,000$ models.

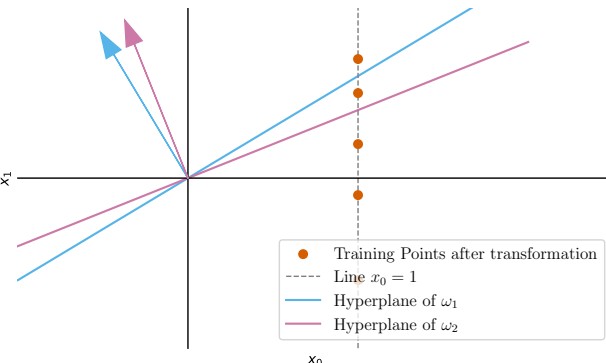

Figure 7: **Visualization of hyperplanes separating training points**. We illustrate how a series of hyperplanes can separate a growing subset of the training points, leading to a triangular, invertible matrix structure as a subset of $\Phi$.

As distribution $\tau(\cdot)$ of the elements $\omega_i \in \mathcal{W}$ we always use $\mathcal{N}(0, I)$. As activation functions, we use ReLU, the Gaussian error function, and the softplus function $\frac{1}{\beta} \cdot \log(1 + \exp(\beta \cdot \omega^\top x))$ with $\beta = 1$. For the first two activation functions, there exist analytically calculatable limiting kernels, the arc-cosine kernel (Cho & Saul, 2009) and the erf-kernel (Williams, 1996). The closed forms for these are

$$k_{\text{arc-cosine}}(x, x') = \frac{1}{2\pi} \|x\| \|x'\| \left(\sin \theta + (\pi - \theta) \cos \theta\right),$$

where $\theta = \cos^{-1}\left(\frac{x^\top x'}{\|x\| \|x'\|}\right)$ and

$$k_{\text{erf}}(x, x') = \frac{2}{\pi} \sin^{-1}\left(\frac{2x^\top x'}{\sqrt{(1 + 2\|x\|^2)(1 + 2\|x'\|^2)}}\right).$$

For the softplus function, we approximate the kernel by estimating the second moment $k(x, x') = \mathbb{E}[\phi(\omega, x)\phi(\omega, x') \mid x, x']$ of the feature extraction using $10^7$ samples from $\tau(\cdot)$. For sampling Gaussian features, we use the same approach as described by Jacot et al. (2020).

Before training on data, we always append a 1 in the zeroeth-dimension of the data before calculating the dot product with $\omega$ (correspondingly, the dimension of $\omega$ is $d + 1$) and applying the activation function. In the ridgeless case, we use $\lambda = 10^{-8}$ to avoid numerical issues.

## A.2. Notes on Stability

During our experiments, we encountered challenges related to both mathematical stability (i.e., matrices being truly singular rather than nearly singular) and numerical stability. This section outlines these issues and describes the steps we took to mitigate them.

Most importantly, the matrix $\Phi_{\mathcal{W}}\Phi_{\mathcal{W}}^\top$ is not almost surely invertible when using the ReLU activation function, meaning that technically, the second condition of our Assumption 2.1 is not fulfilled. In numerical experiments, this results in cases where $(\Phi_{\mathcal{W}}\Phi_{\mathcal{W}}^\top)^{-1}$ is nearly singular (though stabilized with $\lambda = 10^{-8}$).

On the other hand, when $D$ is sufficiently large relative to $N$, $\Phi_{\mathcal{W}}$ is full rank with high probability, which implies that $\Phi_{\mathcal{W}}\Phi_{\mathcal{W}}^\top$ is invertible with high probability. Given our data transformation of appending a 1 in the zeroeth dimension, one can see this as there exists a series of (non-zero probability sets of) hyperplanes separating an increasing subset of the training points, leading to a subset of $\Phi_{\mathcal{W}}$'s columns that form a triangular, invertible matrix (see Fig. 7 for a visualization). Intuitively, higher data dimensionality and better separability of the points increase the probability of $\Phi_{\mathcal{W}}$ having full rank.

As an example of the discussed instabilities, see the adversarial scenario shown in Fig. 8, where $N = 15$ and many training points are placed very close to each other. In this case, individual RF regressors exhibit relatively high variance output values (due to numerical instabilities), which are not averaged out in the "infinite" ensemble. Similar issues were also observed when using the Gaussian error function as the activation function, although they were generally less pronounced.

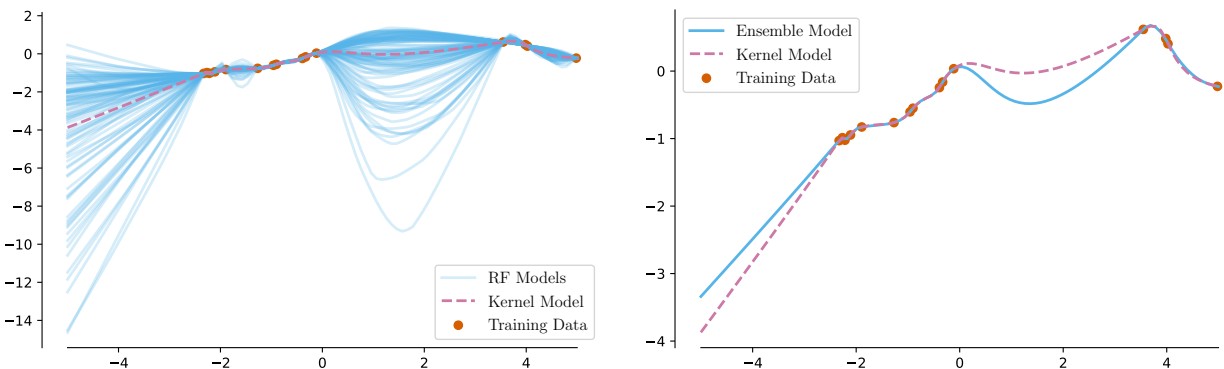

Figure 8: **An adversarial example where the infinite ensemble of overparameterized RF models is numerically not equivalent to a single infinite-width RF model.** (Left) We show a sample of 100 RF models (blue) with ReLU activations trained on the same $N = 15$ densely clustered data points. Additionally, we show the single infinite-width RF model (pink). (Right) We again show the single infinite-width RF model (blue) and the "infinite" ensemble of $M = 10,000$ RF models (pink). A significant difference between the two models is observed in this adversarial case, indicating instability.

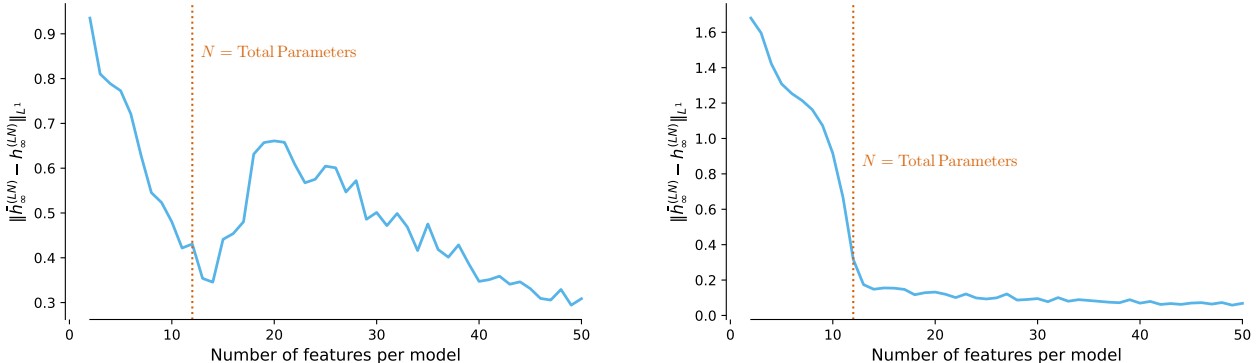

Figure 9: **Using softplus activations instead of ReLU activations reduces instabilities in overparameterized RF ensembles.** The plots show the average absolute difference between the predictions of an infinite ensemble and a single infinite-width model for varying feature counts $D$, using $N = 12$ training samples from the California Housing dataset. (Left) ReLU activations exhibit significant instability, especially for $D > N, D \approx N$, and do not consistently show the expected pointwise equivalence between the infinite ensemble and the single infinite-width model. (Right) Softplus activations — as equivalently shown in Fig. 2 — smooth out these instabilities and more consistently show the expected pointwise equivalence.

To alleviate these issues, we used the following approaches:

- We used a relatively low number of samples, $N = 6$ or $N = 12$, compared to $D = 200$. As shown in Fig. 1, even with $D = 200$, there is still a considerable amount of variance in the RF regressors (i.e., the individual RF regressors are not yet closely approximating the limiting kernel ridge regressor).

- We appended a 1 in the zeroeth dimension of the data before calculating the dot product with $\omega$.

- We performed additional experiments using the softplus function with $\beta = 1$ as a smooth approximation of the ReLU activation function. This often helped stabilize the numerical computations, as seen in Fig. 9, where we repeated a part of the experiment from Fig. 2 using the ReLU function as activation function which increased the numerical instability for low $D$ values.

- We used a ridge term $\lambda = 10^{-8}$ in the ridgeless case to stabilize the inversion of $\Phi_{\mathcal{W}}\Phi_{\mathcal{W}}^{\top}$.

- We used *double precision* for all computations and used the `torch.linalg.lstsq` function with the driver `gelsd` (for not-well-conditioned matrices) to solve linear systems.

- We applied max-min normalization to the entire California Housing dataset to improve stability.

### A.3. Additional Experiments for the Ridgeless Case

To address the question of whether our findings are specific to normally distributed weights $\omega_i$ for the feature generating function, we supplement Fig. 1. Fig. 10 replicates that visualization using weights $\omega_i$ drawn from a Uniform(-10, 10) distribution and the softplus activation function. As can be seen, the equivalence between the infinite ensemble of overparameterized RF models and the single infinite-width RF model remains apparent, and no perceptible difference is observed.

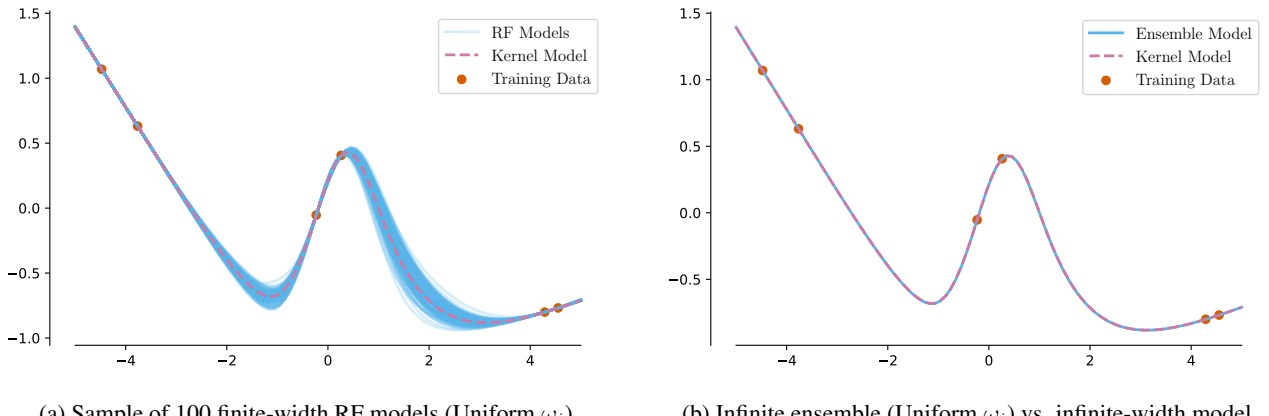

(a) Sample of 100 finite-width RF models (Uniform $\omega_i$)      (b) Infinite ensemble (Uniform $\omega_i$) vs. infinite-width model

Figure 10: **Replication of Fig. 1 with Uniformly Distributed Weights** $\omega_i$**.** Similar to Fig. 1, we demonstrate the equivalence of an infinite ensemble of overparameterized RF models to a single infinite-width RF model. Here, the weights $\omega_i$ for the Softplus activation functions are drawn from a Uniform(-10, 10) distribution. (Left) A sample of 100 finite-width RF models (blue) trained on $N = 6$ data points, with the single infinite-width RF model (pink). (Right) The infinite-width RF model (pink) and the "infinite" ensemble of $M = 10,000$ RF models (blue). No perceptible difference is observed, mirroring the findings with normally distributed weights.

Furthermore, to illustrate the convergence of finite ensembles to the infinite-width model prediction as the number of ensemble members $M$ increases, Fig. 11 expands on the setting of Fig. 1. It shows that even with a small number of ensemble members, the average prediction begins to concentrate around the infinite-width model, and this concentration improves as $M$ grows.

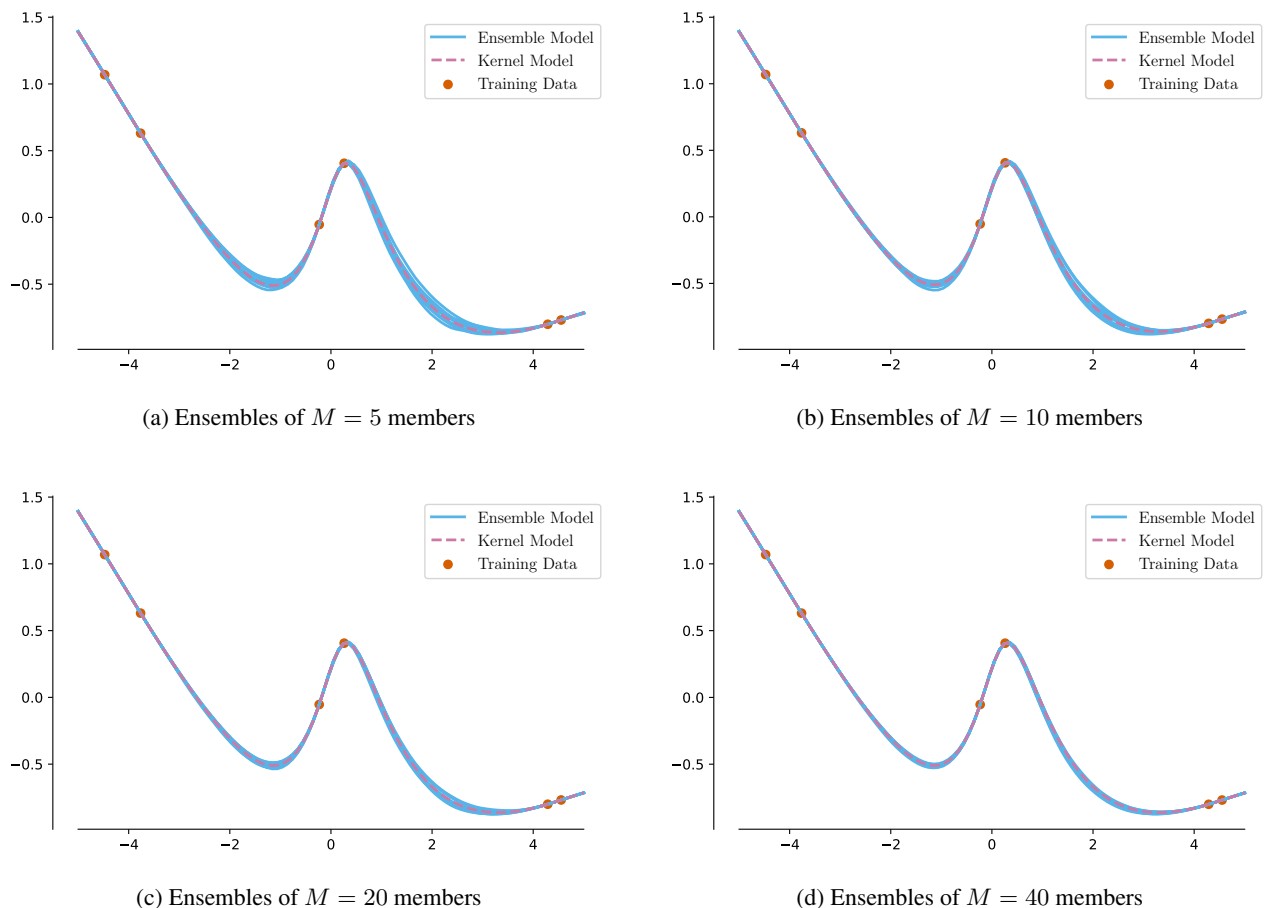

(a) Ensembles of $M = 5$ members

(b) Ensembles of $M = 10$ members

(c) Ensembles of $M = 20$ members

(d) Ensembles of $M = 40$ members

Figure 11: **Evolution of Ensemble Predictions with Increasing Number of Members** ($M$). Following the setup of Fig. 1 (ReLU activations, normally distributed $\omega_i$, $N = 6$ data points), these plots show 10 sample ensemble predictions (blue lines) for varying ensemble sizes $M$. The single infinite-width RF model is shown in pink. As $M$ increases, the ensemble predictions become more concentrated around the infinite-width model.

**Additional experiments on the identity of infinite-width single model and infinite ensembles.** In Fig. 12, we show that the term $\mathbb{E}[w_\perp^\top W^\top (WW^\top)^{-1}]$ is consistently zero for both ReLU and the Gaussian error function activations consistetly with Lemma 3.1. To further demonstrate that this result is not dependent on Gaussian-like weight distributions, Fig. 13 shows this for a softplus activation function, with weights $\omega_i$ drawn from a Uniform(-10, 10) distribution and a Laplace(0, 1) distribution. The expectation of the term remains centered at zero, supporting the generality of our theoretical findings.

**Additional experiments on the ensemble variance.** We observed a different behavior of the RF regressor variance and $r_\perp^2$ as shown in Fig. 4 consistently across different random seeds and dimensions for both ReLU and the Gaussian error function activations as activation functions. In Fig. 14, we present additional examples for the Gaussian error function in one dimension and the ReLU activation in two dimensions.

**Additional experiment on generalization error and variance scaling.** In Fig. 3, the generalization error decay for the ReLU activation function. To verify the consistency of this trend, we repeated the experiment using the Gaussian error function and the corresponding erf-kernel. The result is very similar, shown in Fig. 15. Furthermore, this figure shows that the variance of a single model with $MD$ features decays as $\sim \frac{1}{MD}$, matching the ensemble's behavior.

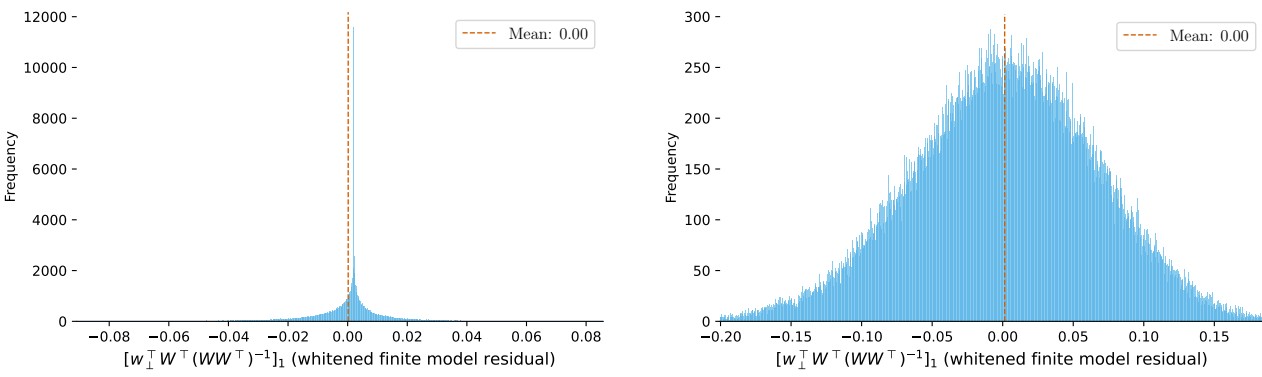

Figure 12: **Empirically, the term $\mathbb{E}[w_\perp^\top W^\top (WW^\top)^{-1}]$ is consistently zero.** We plot the distribution of the first index of $w_\perp^\top W^\top (WW^\top)^{-1}$, which captures the difference between the infinite-width single model and a smaller overparameterized RF model (see Eq. (2)). (Left) We use ReLU as activation function, $x_i \in \mathbb{R}$, and $N = 6, D = 200$. (Right) We use the Gaussian Error activation function, the California Housing dataset and $N = 12, D = 200$.

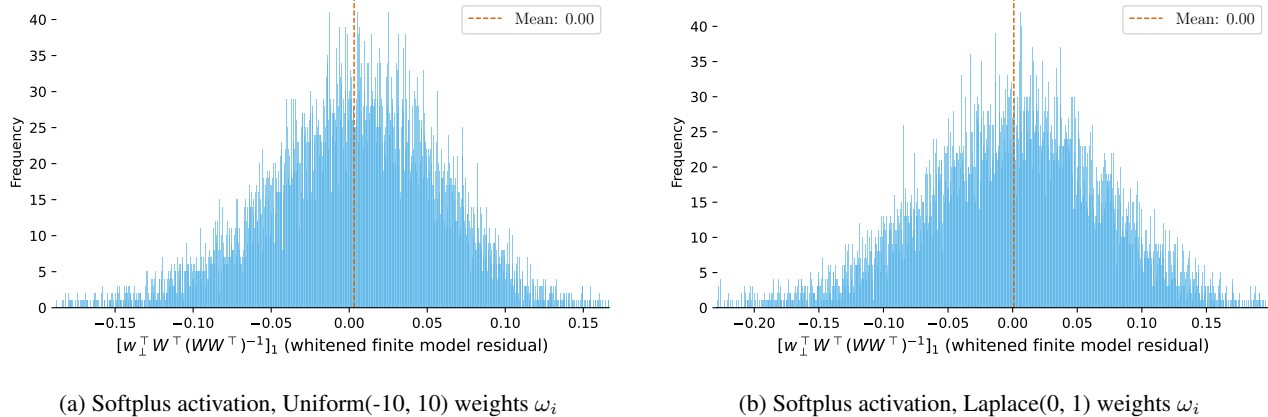

(a) Softplus activation, Uniform(-10, 10) weights $\omega_i$       (b) Softplus activation, Laplace(0, 1) weights $\omega_i$

Figure 13: **Empirical validation of $\mathbb{E}[w_\perp^\top W^\top (WW^\top)^{-1}] \approx 0$ for Softplus activation and non-Gaussian weights.** Similar to Fig. 12, we plot the distribution of the first index of $w_\perp^\top W^\top (WW^\top)^{-1}$. Both plots use a softplus activation function, the California Housing dataset, and $N = 12, D = 200$. (Left) Weights $\omega_i$ are drawn from a Uniform(-10, 10) distribution. (Right) Weights $\omega_i$ are drawn from a Laplace distribution with location 0 and scale 1. In both cases, the distribution is centered at zero.

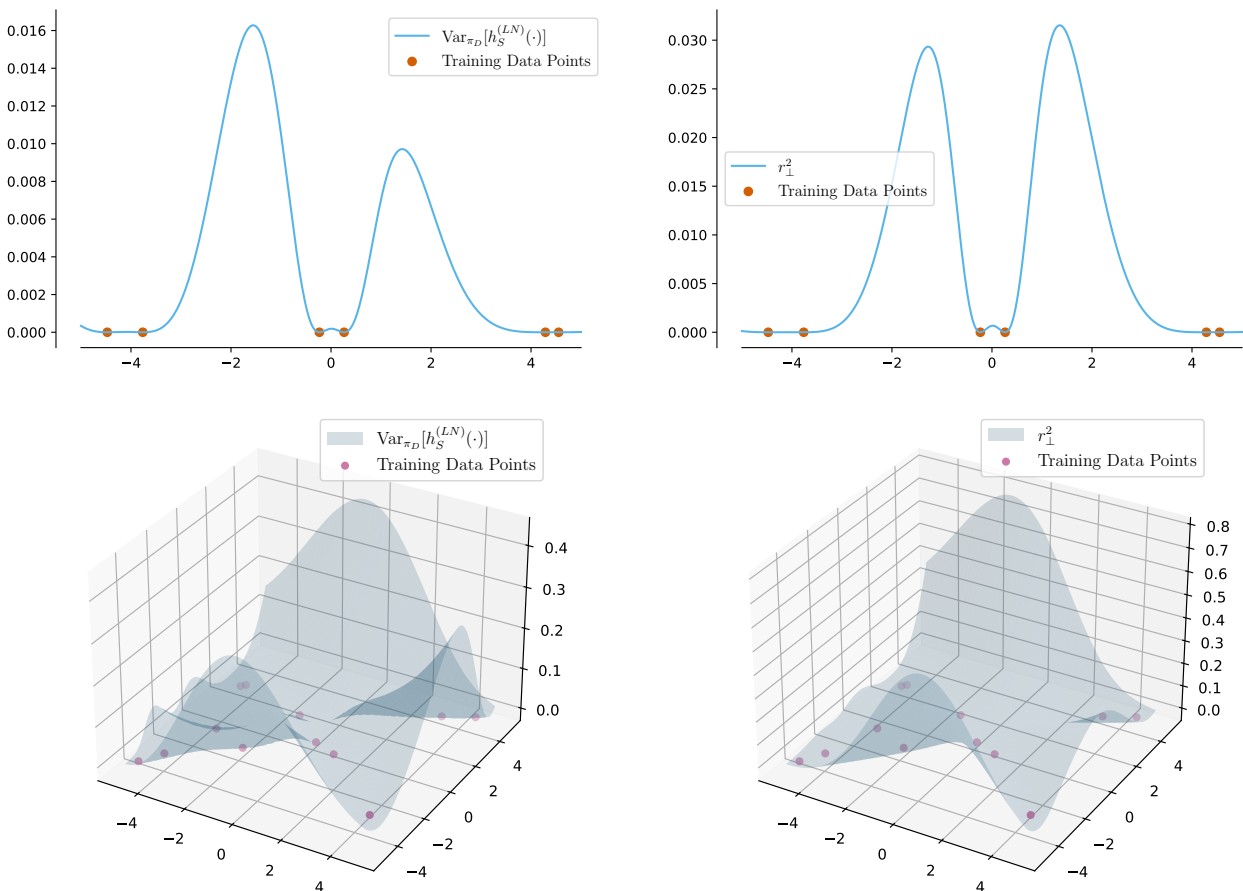

Figure 14: **Variance and $r_\perp^2$ for different activations and dimensions.** (Top left) Variance of RF model predictions across the input range for $D = 200$ and $N = 6$, using the erf activation function. (Top right) Corresponding $r_\perp^2$ values across the input range using the erf kernel. (Bottom left) Variance of RF model predictions across the input range for $D = 200$, $p = 2$, and $N = 12$, using the ReLU activation function. (Bottom right) Corresponding $r_\perp^2$ values across the input range using the arc-cosine kernel.

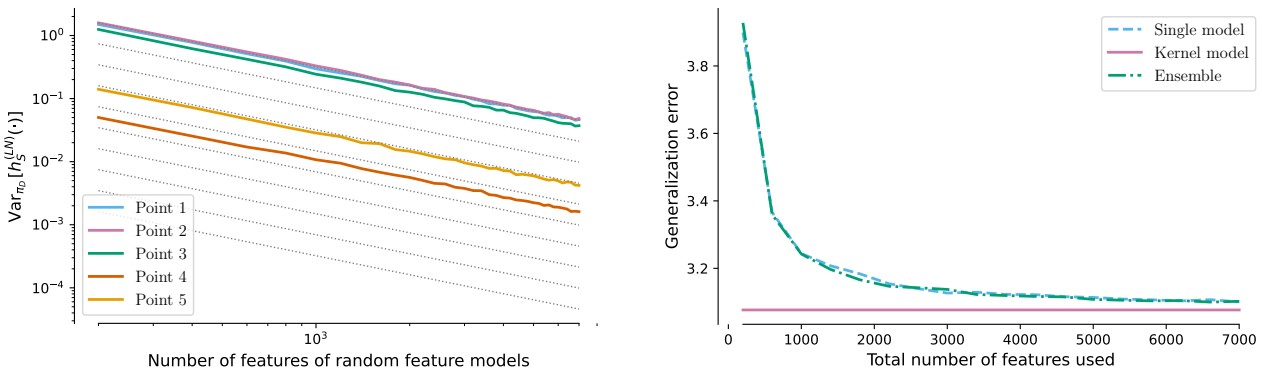

Figure 15: **Variance and generalization error scale similarly with the number of features, consistent with Fig. 3.** In (a), the variance of a single model with $MD$ features decays as $\sim \frac{1}{MD}$, matching the ensemble's behavior. In (b), the generalization error of an ensemble with $M$ models and $D = 200$ features shows a similar decay to that of a single model with $MD$ features. Results use the Gaussian error function, California Housing dataset, and $N = 12$.

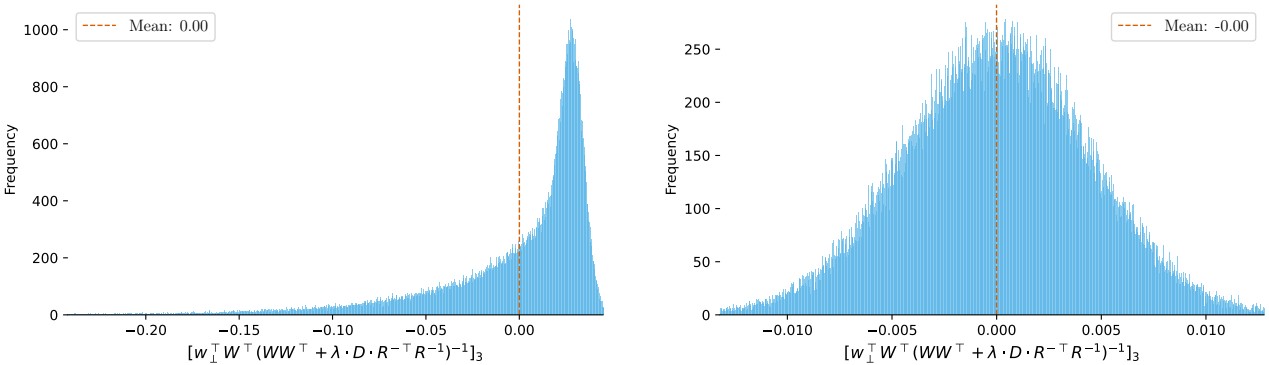

Figure 16: **Empirically, the term** $\mathbb{E}_{W,w_\perp}\left[w_\perp^\top W^\top \left(WW^\top + D \cdot \lambda \cdot R^{-\top}R^{-1}\right)^{-1}\right]$ **is consistently zero.** We show the empirical distribution of an index of $w_\perp^\top W^\top \left(WW^\top + D \cdot \lambda \cdot R^{-\top}R^{-1}\right)^{-1} \in \mathbb{R}^N$, which captures the difference in predictions between $c^\top \mathbb{E}_{W,w_\perp}\left[WW^\top \left(WW^\top + D \cdot \lambda \cdot R^{-\top}R^{-1}\right)^{-1}\right]R^{-\top}y$ and a finite-sized overparameterized RF model (see Eq. (7)). We use $\lambda = 1.0$ in both plots. (Left) We use a ReLU activation function, $x_i \in \mathbb{R}$, and $N = 6, D = 200$. (Right) We use the Gaussian Error Function as activation function, the California Housing dataset, and $N = 12, D = 200$.

## A.4. More Experiments for the Ridge Case

**Additional experiments for the convergence of the expected value term.** In Appx. D, we show that a variant of Lemma 3.1 also holds in the ridge case. More precisely, we show that

$$\mathbb{E}_{W,w_\perp}\left[w_\perp^\top W^\top \left(WW^\top + D \cdot \lambda \cdot R^{-\top}R^{-1}\right)^{-1}\right] = 0$$

under Assumption 2.1. We repeated the experiment from Fig. 12 for the ridge case to verify this experimentally. The results are shown in Fig. 16.

**Additional notes.** In Fig. 5, we illustrate the Lipschitz continuity of the predictions for an infinite ensemble and a kernel regressor with respect to the ridge parameter. Rather than directly presenting the difference $\left|\bar{h}_{\infty,\lambda}^{(RR)}(x^*) - h_{\infty,\lambda}^{(RR)}(x^*)\right|$, we show the evolution of $\left|\bar{h}_{\infty,\lambda}^{(RR)}(x^*) - \bar{h}_{\infty}^{(LS)}(x^*)\right|$ and $\left|h_{\infty,\lambda}^{(RR)}(x^*) - h_{\infty}^{(LS)}(x^*)\right|$. This choice was made because the upper bound we obtained was not consistently tight for settings with large $D$. In particular, the pointwise predictions of the infinite ensemble $\bar{h}_{\infty,\lambda}^{(RR)}$ and the single infinite-width model $h_{\infty,\lambda}^{(RR)}$ trained with ridge $\lambda$ were already very close for non-zero $\lambda$. We opted to display the upper bounds rather than the direct difference to avoid cherry-picking favorable settings.

Our best explanation for this phenomenon is that infinite ensembles under Assumption 2.1 in the ridge regime often behave similarly to the single infinite-width model $h_{\infty,\tilde{\lambda}}^{(RR)}$ with an *implicit ridge* parameter $\tilde{\lambda}$, which solves the equation

$$\tilde{\lambda} = \lambda + \frac{\tilde{\lambda}}{D}\sum_{i=1}^{N}\frac{d_i}{\tilde{\lambda} + d_i}$$

where $d_i$ are the eigenvalues of the kernel matrix $K$, as shown by Jacot et al. (2020) under Gaussianity. Intuitively and empirically, for large $D$, the implicit ridge $\tilde{\lambda}$ tends to be very close to the true ridge $\lambda$. Using Lemma D.1, this suggests that for small values of $\lambda$, the difference between the infinite ensemble and the infinite-width single model $h_{\infty,\lambda}^{(RR)}$ with ridge $\lambda$ is already minimal before $\lambda$ approaches zero.

Interestingly, our findings (see Fig. 2) suggest that in the ridgeless case, the similarity to the ridge regressor with the implicit ridge only holds in the overparameterized regime. Note that this does not violate the results from Jacot et al. (2020) since the constants in their bounds blow up as $\lambda \to 0$ in both the underparameterized and overparameterized regimes.

## B. Experimental Setup and Additional Results for Neural Network Models

### B.1. Experimental Setup

For all our experiments with neural networks, we used a three-layer MLP with hidden layers of equal width and ReLU activations. Models were trained for 1000 epochs using SGD with momentum, a learning rate of $0.01$, and a momentum decay of $0.9$.

Training was performed on the same set of 12,000 samples from the California Housing dataset, with a validation set of 3,000 samples and a test set of 5,000 samples. Since the number of parameters scales quadratically with the hidden layer width, the overparameterized regime is reached at a width of approximately 80.

All reported results are based on the best checkpoint selected using validation performance over the 1000 training epochs.

### B.2. Additional Results and Limitations of Our Experiments

The results in Fig. 2, which show the average absolute difference in predictions between large ensembles with increasing number of parameters ($D$) in the component models and a single large model (with $MD$ parameters), are further supported by the increasing correlation of residuals between the ensemble and the single model, as shown in Fig. 17. When the component models of the ensemble become overparameterized, ensemble residuals align better with those of a single large model, indicating that overparameterized ensembles make more similar errors to a large single model than their underparameterized counterparts.

Furthermore, the correlations of the residuals of large ensembles of overparameterized models and two large single models trained with different initializations were comparably high, with the average correlation between an ensemble and a single model even slightly higher. At the same time, the residual correlation between two large ensembles trained with different initializations was significantly higher than both of these correlations. The lower variance in predictions across multiple ensembles compared to a single large model does not align with our theoretical expectations and experiments with random feature models. We hypothesize that this discrepancy arises from large single models being more unstable to train but did not investigate this limitation of our experiments in more detail.

## C. Proofs for Overparameterized Ridgeless Regression

### C.1. Equivalence of Infinite Ensemble and Infinite Single Model

We start by proving the equivalent formulation of the infinite ensemble prediction stated in Eq. (2) using the terms $W$ and $w_\perp$ as introduced in Sec. 2:

*Proof.* Defining $\phi^*_{\mathcal{W}} = [\phi(\omega_i, x^*)]_i \in \mathbb{R}^D$, we have

$$
\begin{aligned}
\bar{h}_\infty(x^*) &= \mathbb{E}_{\mathcal{W}} \left[ \tfrac{1}{D} \phi^*_{\mathcal{W}} \Phi^\top_{\mathcal{W}} \left( \tfrac{1}{D} \cdot \Phi_{\mathcal{W}} \Phi^\top_{\mathcal{W}} \right)^{-1} \right] y \\
&= \mathbb{E}_{W,w_\perp} \left[ \left( c^\top W + r_\perp w_\perp^\top \right) W^\top R \left( R^\top W W^\top R \right)^{-1} \right] y \\
&= \mathbb{E}_{W,w_\perp} \left[ \left( c^\top W + r_\perp w_\perp^\top \right) W^\top \left( W W^\top \right)^{-1} \right] R^{-\top} y \\
&= c^\top R^{-\top} y \ + \ r_\perp \mathbb{E}_{W,w_\perp} \left[ w_\perp^\top W^\top \left( W W^\top \right)^{-1} \right] R^{-\top} y,
\end{aligned}
\tag{6}
$$

where $c, R, r_\perp$ are as defined in Eq. (1). The left term in Eq. (6) is equal to $h^{(\mathrm{LN})}_\infty(x)$:

$$
h^{(\mathrm{LN})}_\infty(x^*) = \left[ k(x_i, x^*) \right]^N_{i=1} K^{-1} y = c^\top R R^{-1} R^{-\top} y = c^\top R^{-\top} y.
$$

$\square$

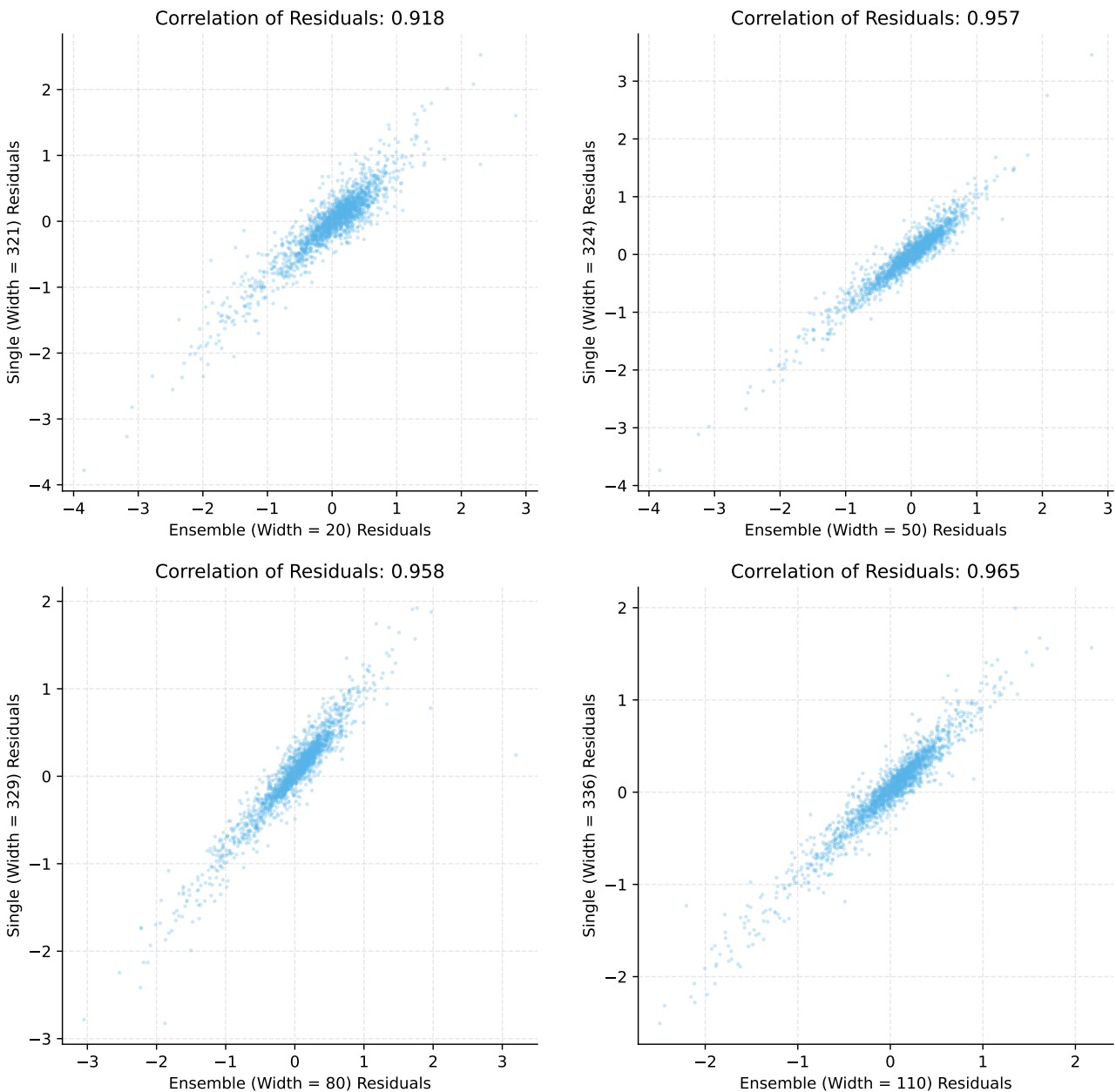

Figure 17: **Correlation of residuals between ensembles and a single large model.** Scatter plots comparing the residuals of a single large model with $MD$ parameters to those of ensembles with increasing component parameters (shown by the the width of the component models): (Top left) 20, (Top right) 50, (Bottom left) 80, and (Bottom right) 110. The correlation of residuals increases as the component parameter count grows. This suggests that overparameterized ensembles make more similar errors to a single large model than (strongly) underparameterized ensembles.

In the case of $\lambda > 0$, we can similarly see that

$$
\begin{aligned}
\bar{h}_{\infty,\lambda}^{(RR)}(x^*) &= \mathbb{E}_{\mathcal{W}}\left[\tfrac{1}{D}\phi_{\mathcal{W}}^*\Phi_{\mathcal{W}}^\top\left(\tfrac{1}{D}\cdot\Phi_{\mathcal{W}}\Phi_{\mathcal{W}}^\top + \lambda I\right)^{-1}\right]y \\
&= \mathbb{E}_{W,w_\perp}\left[\left(c^\top W + r_\perp w_\perp^\top\right)W^\top R\left(R^\top W W^\top R + D\cdot\lambda\cdot R^\top R^{-\top}R^{-1}R\right)^{-1}\right]y \\
&= \mathbb{E}_{W,w_\perp}\left[\left(c^\top W + r_\perp w_\perp^\top\right)W^\top\left(W W^\top + D\cdot\lambda\cdot R^{-\top}R^{-1}\right)^{-1}\right]R^{-\top}y \\
&= c^\top\mathbb{E}_{W,w_\perp}\left[W W^\top\left(W W^\top + D\cdot\lambda\cdot R^{-\top}R^{-1}\right)^{-1}\right]R^{-\top}y \\
&\quad + r_\perp\mathbb{E}_{W,w_\perp}\left[w_\perp^\top W^\top\left(W W^\top + D\cdot\lambda\cdot R^{-\top}R^{-1}\right)^{-1}\right]R^{-\top}y.
\end{aligned}
\tag{7}
$$

Note that the simplification demonstrated in Eq. (2) does not work as nicely in the underparameterized case ($D \leq N$). This is because the weights, in this case, are given by $\theta = (\Phi_{\mathcal{W}}^\top\Phi_{\mathcal{W}})^{-1}\Phi_{\mathcal{W}}^\top y$, and thus the infinite ensemble prediction expands as:

$$
\begin{aligned}
\bar{h}_\infty(x^*) &= \mathbb{E}_{\mathcal{W}}\left[\phi_{\mathcal{W}}^*\left(\Phi_{\mathcal{W}}^\top\Phi_{\mathcal{W}}\right)^{-1}\Phi_{\mathcal{W}}^\top\right]y \\
&= \mathbb{E}_{W,w_\perp}\left[\left(c^\top W + r_\perp w_\perp^\top\right)\left(W^\top R R^\top W\right)^{-1}W^\top R\right]y.
\end{aligned}
$$

Here, $RR^\top$ lies inside the inverse, preventing the simplifications available in the overparameterized regime.

Next up, we show that the expected value $\mathbb{E}_{W,w_\perp}\left[w_\perp^\top W^\top\left(W W^\top\right)^{-1}\right]$ is zero under Assumption 2.1. This directly implies the pointwise equivalence of the infinite ensemble and the single infinite-width model (see Theorem 3.2).

*Lemma 3.1 (Restated).* Under Assumption 2.1, it holds that $\mathbb{E}_{W,w_\perp}[w_\perp^\top W^\top(W W^\top)^{-1}] = 0$.

*Proof.* Define $A_{-i} = (W W^\top - w_i w_i^\top)$. Note that $A_{-1}$ is almost surely invertible and positive definite by assumption Assumption 2.1.

By the Woodbury formula, for almost every $W W^\top$ we have that

$$
(W W^\top)^{-1} = (A_{-i} + w_i w_i^\top)^{-1} = A_{-i}^{-1} - \frac{A_{-i}^{-1}w_i w_i^\top A_{-i}^{-1}}{1 + w_i^\top A_{-i}^{-1}w_i},
$$

which implies that

$$
\begin{aligned}
w_\perp^\top W^\top(W W^\top)^{-1} &= \sum_{i=1}^{D} w_{\perp i}w_i^\top\left(A_{-i}^{-1} - \frac{A_{-i}^{-1}w_i w_i^\top A_{-i}^{-1}}{1 + w_i^\top A_{-i}^{-1}w_i}\right) \\
&= \sum_{i=1}^{D} w_{\perp i}\left(w_i^\top\frac{A_{-i}^{-1} + w_i^\top A_{-i}^{-1}w_i^\top A_{-i}^{-1}w_i}{1 + w_i^\top A_{-i}^{-1}w_i} - \frac{w_i^\top A_{-i}^{-1}w_i w_i^\top A_{-i}^{-1}}{1 + w_i^\top A_{-i}^{-1}w_i}\right) \\
&= \sum_{i=1}^{D}\frac{w_{\perp i}w_i^\top}{1 + w_i^\top A_{-i}^{-1}w_i}A_{-i}^{-1}.
\end{aligned}
$$

For any positive definite matrix $B \in \mathbb{R}^{N\times N}$ and any vector $v \in \mathbb{R}^N$; $\|v\| = 1$ and any $i \in \{1,...,D\}$, we have

$$
\begin{aligned}
\left|\mathbb{E}_{w_{\perp i},w_i}\left[\frac{w_{\perp i}w_i^\top}{1 + w_i^\top B w_i}\right]v\right| &\leq \mathbb{E}_{w_{\perp i},w_i}\left[\left|\frac{w_{\perp i}w_i^\top v}{1 + w_i^\top B w_i}\right|\right] \\
&= \int_0^\infty \mathbb{P}\left[\left|\frac{w_{\perp i}w_i^\top v}{1 + w_i^\top B w_i}\right| \geq t\right]dt \\
&= \int_0^\infty \mathbb{P}\left[\left|w_{\perp i}w_i^\top v\right| \geq \left(1 + w_i^\top B w_i\right)t\right]dt \\
&\leq \int_0^\infty \mathbb{P}\left[\left|w_{\perp i}w_i^\top\right| > t\right]dt \\
&\leq \int_0^{\nu^2/\alpha} 2\exp\left(-\frac{t^2}{2\nu}\right)dt + \int_{\nu^2/\alpha}^\infty 2\exp\left(-\frac{t}{2\alpha}\right)dt,
\end{aligned}
\tag{8}
$$

where the last inequality is a standard sub-exponential bound applied to $w_{\perp i} w_i$. Note that we here use the fact that $\mathbb{E}[w_{\perp i} w_i^\top] = 0$ and the $(\nu^2, \alpha)$-sub-exponentiality of $\left| w_{\perp i} w_i^\top \right|$.

Since the last two integrals in Eq. (8) are finite, the expectation $\mathbb{E}_{W, w_\perp} \left[ (w_{\perp i} w_i^\top)/(1 + w_i^\top B w_i) \right] v$ is finite. By the weak law of large numbers, for i.i.d. random variables $w_i^{(j)}$ and $w_{\perp i}^{(j)}$ across different $j$'s, we have

$$\mathbb{P} \left[ \left| \frac{1}{M} \sum_{j=1}^M \frac{w_{\perp i}^{(j)} (w_i^{(j)})^\top v}{1 + (w_i^{(j)})^\top B w_i^{(j)}} - \mathbb{E}_{W, w_\perp} \left[ \frac{w_{\perp i} w_i^\top}{1 + w_i^\top B w_i} \right] v \right| > t \right] \to 0,$$

for any $t > 0$ and $v \in \mathbb{R}^N$ such that $\|v\| = 1$ as $M \to \infty$. At the same time, repeating the sub-exponential argument above, we have that

$$\mathbb{P} \left[ \left| \frac{1}{M} \sum_{j=1}^M \frac{w_{\perp i}^{(j)} (w_i^{(j)})^\top v}{1 + (w_i^{(j)})^\top B w_i^{(j)}} \right| > t \right] \leq \mathbb{P} \left[ \left| \frac{1}{M} \sum_{i=1}^M w_{\perp i}^{(j)} (w_i^{(j)})^\top v \right| > t \right]$$

$$\leq \begin{cases} 2 \exp \left( -\frac{M t^2}{2\nu} \right) & 0 < t \leq \nu^2/\alpha \\ 2 \exp \left( -\frac{M t}{2\alpha} \right) & t > \nu^2/\alpha \end{cases}$$

$$\to 0$$

as $M \to \infty$. Here we use the property that the sum of $M$ $(\nu^2, \alpha)$-sub-exponential random variables is $(M\nu^2, \alpha)$-sub-exponential.

Together, these results imply that $\mathbb{E}_{W, w_\perp} \left[ (w_{\perp i} w_i^\top)/(1 + w_i^\top B w_i) \right] = 0$ for every positive definite $B$. Since the random matrix $A_{-i}$ is positive semidefinite, almost surely invertible (by the second half of Assumption 2.1), and independent of $w_i, w_{\perp i}$, we have that

$$\mathbb{E}_{w_\perp, W} \left[ w_\perp W^\top \left( W W^\top \right)^{-1} \right] = \sum_{i=1}^D \mathbb{E}_{w_{\perp i}, w_i, A_{-i}} \left[ \frac{w_{\perp i} w_i^\top}{1 + w_i^\top A_{-i}^{-1} w_i} A_{-i}^{-1} \right]$$

$$= \sum_{i=1}^D \mathbb{E}_{A_{-i}} \left[ \mathbb{E}_{w_{\perp i}, w_i} \left[ \frac{w_{\perp i} w_i^\top}{1 + w_i^\top A_{-i}^{-1} w_i} \right] A_{-i}^{-1} \right] = 0.$$

$\square$

We remark that this proof equivalently holds for the ridge-regression case, i.e., $\mathbb{E}_{W, w_\perp} \left[ w_\perp^\top W^\top \left( W W^\top + D \cdot \lambda \cdot R^{-\top} R^{-1} \right)^{-1} \right] = 0$ since the proof does not rely on the specific form of the matrix $A_{-i}$ other than it being positive definite. Thus by Eq. (7) we directly get that under Assumption 2.1 it holds that

$$\bar{h}_{\infty, \lambda}^{(RR)}(x^*) = c^\top \mathbb{E}_{W, w_\perp} \left[ W W^\top \left( W W^\top + D \cdot \lambda \cdot R^{-\top} R^{-1} \right)^{-1} \right] R^{-\top} y. \tag{9}$$

### C.2. Ensembles versus Larger Single Models under a Finite Feature Budget

We now prove the formal version of Theorem 3.3.

Let's first restate the informal version of the theorem:

*Theorem 3.3 (Restated).* Under slightly stronger assumptions than Assumption 2.1, the $L_2$ difference between a single neural network with $MD$ features and an ensemble of $M$ neural networks each with $D$ features is, with probability $1 - \delta$, upper bounded by:

$$\left\| h_{\mathcal{W}^*}^{(LN)}(\cdot) - \bar{h}_{\mathcal{W}_{1:M}}^{(LN)}(\cdot) \right\|_2^2 \leq O(\sqrt{\log(1/\delta)}) + O(1/D)$$

We now provide the formal version of this theorem, which uses the following definitions:

**Definition C.1.** Define $\Sigma : \mathcal{L}^2(\mathcal{X}) \to \mathcal{L}^2(\mathcal{X})$ as $\Sigma f = \int_{\mathcal{X}} k(x, \cdot) f(x) d\mu(x)$.

**Definition C.2.** For any fixed set of random features $\mathcal{W} = \{\omega_1, \ldots, \omega_D\}$ of size $D$ define $\phi_{\mathcal{W}}(x) = \frac{1}{\sqrt{D}} [\phi(\omega_1, x), \ldots, \phi(\omega_D, x)]^\top$, and the approximated kernel function $\hat{k}_{\mathcal{W}}(x, \cdot) = \phi_{\mathcal{W}}(x)^\top \phi_{\mathcal{W}}(\cdot)$. Using this, define $\hat{\Sigma}_{\mathcal{W}} : \mathcal{L}^2(\mathcal{X}) \to \mathcal{L}^2(\mathcal{X})$ as $\hat{\Sigma}_{\mathcal{W}} f = \int_{\mathcal{X}} \hat{k}_{\mathcal{W}}(x, \cdot) f(x) d\mu(x)$.

(We will drop the $\mathcal{W}$ subscript from $\phi_{\mathcal{W}}$, $\hat{k}_{\mathcal{W}}$, and $\hat{\Sigma}_{\mathcal{W}}$ when the set of random features is clear from context.) Now, we state the assumptions that we need for the proof which are stronger than the assumptions in Assumption 2.1:

**Assumption C.3.** We make the following assumptions:

- The columns of $\Phi$ are subgaussian with constant $L$, i.e. $P(|Xv| \geq t) \leq 2\exp\left(-t^2/L^2\right)$ for all $v \in \mathbb{R}^D$ with $|v| \leq 1$.

- For any $\delta_1 \in (0,1)$ there exists a $C < \infty$ and $D_0(\delta_1)$ such that for all $D \geq D_0(\delta_1)$ it holds that $\left\|\hat{\Sigma} - \Sigma\right\| \leq C$ with probability $\geq 1 - \delta_1$.

- The feature extraction $\phi(\omega, \cdot)$ is almost surely square integrable over the data probability measure (i.e. $\mathbb{E}_x[\phi(\omega, \cdot)^2] < \infty$).

**Theorem C.4** (Non-asymptotic bound on the $L_2$ difference between ensembles and single models (informal version)). *Under Assumption C.3, there exist constants $c_1, c_2, c_3$ such that for any $\delta_1 \in (0,1)$ and all $M, N, D$ with $M \cdot D \geq D_0(\delta_1)$ and defining $\lambda_{\min} := \min(1, \lambda_{\min}(K))$ it holds:*

*If $\frac{\lambda_{\min}}{2 \cdot L^2} - \frac{c_1}{L^2}\left\{\sqrt{\frac{N}{D}} + \frac{N}{D}\right\} > 0$ and $\delta_2 = M \cdot c_2 e^{-c_3 D \min\left(\kappa_{2j}, \kappa_{2j}^2\right)} + c_2 e^{-c_3 MD \min\left(\kappa_1, \kappa_1^2\right)} < 1 - \delta_1$, where $\kappa_1 = \max(0, \frac{\lambda_{min}}{2 \cdot L^2} - \frac{c_1}{L^2}\left\{\sqrt{\frac{N}{M \cdot D}} + \frac{N}{M \cdot D}\right\})$ and $\kappa_{2j} = \max(0, \frac{\lambda_{\min}}{2 \cdot L^2} - \frac{c_1}{L^2}\left\{\sqrt{\frac{N}{D}} + \frac{N}{D}\right\})$, then for any $\delta_3 \in (0, 1 - \delta_1 - \delta_2)$ it holds with probability at least $1 - \delta_1 - \delta_2 - \delta_3$ that the $L_2$-norm of the difference between the larger, but finite-width single model and the finite ensemble with the same features is bounded by*

$$\left\|h_{\mathcal{W}}^{(\text{LN})}(\cdot) - \bar{h}_{W_{1,M}}(\cdot)\right\|_2^2 \leq \epsilon + O(1/D)$$

*where $\epsilon = \sqrt{\frac{1}{\lambda_{min}} \log\left(\frac{2}{\delta_3}\right)}$.*

*Proof.*

***First step: Expressing as the difference in their parameter norms.***

In the following, we define $\phi(x) = \frac{1}{\sqrt{MD}}[\phi(\omega_1, x), \ldots, \phi(\omega_{MD}, x)]^\top$. Using this definition we get that with $\theta^{(\text{ENS})} = \frac{1}{\sqrt{M}}[\theta_1^{(\text{ENS})}, \ldots, \theta_M^{(\text{ENS})}]^\top$, where $\theta_j^{(\text{ENS})}$ are the parameters of the $j$-th component model, we get $\phi(x)^\top \theta^{(\text{ENS})} = \frac{1}{M}\sum_{j=1}^{M}\frac{1}{\sqrt{D}}[\phi(\omega_{(j-1)D}, x), \ldots, \phi(\omega_{jD}, x)]^\top \theta_j^{(\text{ENS})}$. At the same time, we can write $\phi(x)^\top \theta^{(\text{Single})} = \frac{1}{\sqrt{MD}}[\phi(\omega_1, x), \ldots, \phi(\omega_{MD}, x)]^\top \theta^{(\text{Single})}$.

Using an equivalence of norm argument, we get $\mathbb{E}_{\boldsymbol{x}}\left[\left(\phi(\boldsymbol{x})^\top \theta^{(\text{ENS})} - \phi(\boldsymbol{x})^\top \theta^{(\text{Single})}\right)^2\right] \leq C_{\text{upper}}\left\|\theta^{(\text{ENS})} - \theta^{(\text{Single})}\right\|_2^2$.

More precisely, for this argument, $C_{\text{upper}}$ will be the operator norm of the matrix $\mathbb{E}_{\boldsymbol{x}}[\phi(\boldsymbol{x})\phi(\boldsymbol{x})^\top]$. Since this is a positive semi-definite matrix, the operator norm is equal to its largest eigenvalue.

We can now define the operator $T : L^2(\mathcal{X}) \to \mathbb{R}^D$, $\quad T(f) = \int_{\mathcal{X}} \phi(x)f(x)d\mu(x)$. The adjoint operator is $T^* : \mathbb{R}^D \to L^2(\mathcal{X})$, $\quad T^*(y) = \sum_{i=1}^{D} y_i \phi(\omega_i, \cdot)$.

Let's see how $TT^*$ acts on a vector $v \in \mathbb{R}^D$:

$$\begin{aligned}
TT^*v &= T(T^*v) \\
&= T\left(\phi(\cdot)^\top v\right) \\
&= \int_{\mathcal{X}} \phi(x)\phi(\cdot)^\top v d\mu(x) \\
&= \int_{\mathcal{X}} \phi(x)\phi(\cdot)^\top d\mu(x)v \\
&= E_{\boldsymbol{x}}[\phi(\boldsymbol{x})\phi(\boldsymbol{x})^\top]v
\end{aligned}$$

Thus, we have that $TT^* = E_{\boldsymbol{x}}[\phi(\boldsymbol{x})\phi(\boldsymbol{x})^\top]$. We know from linear algebra that $TT^*$ has the same eigenvalues as $T^*T$. Thus, it is enough to bound the eigenvalues of $\hat\Sigma = T^*T$.

To bound the eigenvalues of $\hat\Sigma$, we bound its difference in operator norm to the equivalent operator for the true kernel $K$, i.e. $\Sigma f = \int_{\mathcal{X}} k(x,\cdot)f(x)d\mu(x)$. Since we assume that for $MD > D_0(\delta_1)$ we have that $\left\|\hat\Sigma - \Sigma\right\| \leq C$ with probability $\geq 1 - \delta_1$, we get that $\left\|\hat\Sigma\right\| \leq \|\Sigma\| + C$ with probability $\geq 1 - \delta_1$. This implies that with probability $\geq 1 - \delta_1$ $C_{\text{upper}}$ is bounded by a constant independent of $M$ and $D$.

*Second step: Using least norm geometry.*

By least norm geometry, we get that $\left\|\theta^{(\text{ENS})} - \theta^{(\text{Single})}\right\|_2^2 = \left\|\theta^{(\text{ENS})}\right\|_2^2 - \left\|\theta^{(\text{Single})}\right\|_2^2$. Furthermore, we directly get that the norm of the ensemble as the sum of the norms of the component models, i.e. $\left\|\theta^{(\text{ENS})}\right\|_2^2 = \frac{1}{M}\sum_{j=1}^M \left\|\theta_j^{(\text{ENS})}\right\|_2^2$.

*Third step: bound the probability that all empirical kernel inverses admit a Taylor expansion and have bounded lower eigenvalues.*

We now want to bound probability that the eigenvalues of $\frac{1}{M*D}\Phi\Phi^T - K$—i.e. the difference between the empirical kernel matrix and the true kernel matrix—are bigger than $\frac{\lambda_{min}}{2}$. If the eigenvalues are less than $\frac{\lambda_{min}}{2}$, then

1. the inverse of the matrix $K^{-1/2}\frac{1}{M*D}\Phi\Phi^T K^{-1/2}$ admits a Taylor expansion and

2. the minimum eigenvalue of $(\frac{1}{MD}\Phi\Phi^T)^{-1}$ will be lower bounded.

We use the following concentration inequality to bound this probability:

**Lemma C.5** ([Wainwright (2019)](#), Thm. 6.5)**.** *Let* $x_1, \ldots, x_n \in \mathbb{R}^d$ *be i.i.d. $L$-subgaussian random variables with* $A = \mathbb{E}[x_i x_i^\top] \in \mathbb{R}^{d\times d}$*. Then for any $\delta \geq 0$, there exists some $c_1, c_2, c_3 > 0$ so that*

$$\mathbb{P}\left[\frac{\|\frac{1}{n}\sum_{i=1}^n x_i x_i^\mathrm{T} - A\|_2}{L^2} \geq c_1\left\{\sqrt{\frac{d}{n}} + \frac{d}{n}\right\} + \delta\right] \leq c_2 e^{-c_3 n \min(\delta,\delta^2)}$$

Applying Lemma C.5 to the case of the random matrix $\frac{1}{M\cdot D}\Phi\Phi^T - K$, we have that $n = M*D$, $d = N$, $A = K$ and $x_i$ is the $i$-th column of $\Phi$. Furthermore, we want to bound the probability that $\frac{\|\frac{1}{M\cdot D}\Phi\Phi^T - K\|_2}{L^2}$ is bigger than $\frac{\lambda_{min}}{2\cdot L^2}$. Thus, we set $\kappa_1 = \max(0, \frac{\lambda_{min}}{2\cdot L^2} - \frac{c_1}{L^2}\left\{\sqrt{\frac{N}{M\cdot D}} + \frac{N}{M\cdot D}\right\})$ and get that if $\frac{\lambda_{min}}{2\cdot L^2} - \frac{c_1}{L^2}\left\{\sqrt{\frac{N}{M\cdot D}} + \frac{N}{M\cdot D}\right\} > 0$, then the probability that the eigenvalues of the difference are bigger than $\frac{\lambda_{min}}{2}$ is at most $c_2 e^{-c_3 M\cdot D \min(\kappa_1, \kappa_1^2)}$.

Similarly, we get that the probability that the eigenvalues of the difference for a single component model $(\frac{1}{D}\Phi_j \Phi_j^T - K)$ are bigger than $\frac{\lambda_{min}}{2}$ is bounded by $c_2 e^{-c_3 D \min(\kappa_{2j}, \kappa_{2j}^2)}$, where we define $\kappa_{2j} = \max(0, \frac{\lambda_{min}}{2*L^2} - \frac{c_1}{L^2}\left\{\sqrt{\frac{N}{D}} + \frac{N}{D}\right\})$. The probability that the eigenvalues of any of the component models are bigger than $\frac{\lambda_{min}}{2}$ is then by a union bound bounded by $M \cdot c_2 e^{-c_3 D \min(\kappa_{2j}, \kappa_{2j}^2)}$ (again if $\frac{\lambda_{min}}{2\cdot L^2} - \frac{c_1}{L^2}\left\{\sqrt{\frac{N}{D}} + \frac{N}{D}\right\} > 0$).

We now define $\delta_2 = M \cdot c_2 e^{-c_3 D \min(\kappa_{2j}, \kappa_{2j}^2)} + c_2 e^{-c_3 D \min(\kappa_1, \kappa_1^2)}$.

*Fourth step: assume that all empirical kernel matrices come from a truncated distribution.*

Let $\tilde\pi_D$ and $\tilde\pi_{MD}$ be the distributions over $\frac{1}{D}\Phi_i\Phi_i^\top$ and $\frac{1}{MD}\Phi\Phi^\top$ matrices *conditioned* on the fact that their inverse matrices admit a Taylor expansion. With probability $1 - \delta_2$, the $\frac{1}{D}\Phi_i\Phi_i^\top$ matrices that form our ensemble and single model admit Taylor expansions. In other words, with probability $1 - \delta_2$ we can view the $\frac{1}{D}\Phi_i\Phi_i^\top$ matrices as i.i.d. draws from $\tilde\pi_D$ and we can view $\frac{1}{MD}\Phi\Phi^\top$ as a draw from $\tilde\pi_D$.

*Fifth step: bound the difference between the expected ensemble and single model inverses under the truncated distribution.*

Under $\tilde{\pi}_D$, we have that:

$$\mathbb{E}_{\tilde{\pi}_D}\left[K^{1/2}(\tfrac{1}{D}\Phi_i\Phi_i^\top)^{-1}K^{1/2}\right] = \mathbb{E}_{\tilde{\pi}_D}\left[\left(I - \left(I - K^{-1/2}(\tfrac{1}{D}\Phi_i\Phi_i^\top)K^{-1/2}\right)\right)^{-1}\right]$$

$$= \sum_{i=0}^{\infty}\mathbb{E}_{\tilde{\pi}_D}\left[\left(I - K^{-1/2}(\tfrac{1}{D}\Phi_i\Phi_i^\top)K^{-1/2}\right)^i\right]$$

The zero-th term in the Taylor expansion is $I$. Recognizing that $(\tfrac{1}{D}\Phi_i\Phi_i^\top)$ is a sample mean random feature outer products with expectation $K$, the first term in the Taylor expansion is 0. Using formula (15) from (Angelova, 2012) we find that the second term is equal to $\tfrac{1}{D}M_2$ for some constant $M_2$ an all other terms are $O(\tfrac{1}{D^2})$. Thus:

$$\mathbb{E}_{\tilde{\pi}_D}\left[K^{1/2}(\tfrac{1}{D}\Phi_i\Phi_i^\top)^{-1}K^{1/2}\right] = I + \frac{1}{D}M_2 + O\left(\frac{1}{D^2}\right).$$

Following the same argument we have that

$$\mathbb{E}_{\tilde{\pi}_{MD}}\left[K^{1/2}(\tfrac{1}{MD}\Phi\Phi^\top)^{-1}K^{1/2}\right] = I + \frac{1}{MD}M_2 + O\left(\frac{1}{(MD)^2}\right).$$

Thus,

$$\frac{1}{M}\sum_{i=1}^{M}\mathbb{E}_{\tilde{\pi}_D}\left[y^\top\left(\frac{1}{D}\Phi_i\Phi_i^\top\right)^{-1}y\right] - \mathbb{E}_{\tilde{\pi}_{MD}}\left[y^\top\left(\frac{1}{MD}\Phi\Phi^\top\right)^{-1}y\right]$$

$$= y^\top K^{-1/2}\left(\left[I + \frac{1}{D}M_2 + O\left(\frac{1}{D^2}\right)\right] - \left[I + \frac{1}{MD}M_2 + O\left(\frac{1}{(MD)^2}\right)\right]\right)K^{-1/2}y$$

$$= (y^\top K^{-1}y)\, O\left(\frac{1}{D}\right)$$

### Sixth step: Using a Hoeffding bound.

Lastly, we bound the difference between $y(\tfrac{1}{MD}\Phi\Phi^\top)^{-1}y$ and its expected value (over the truncated distribution). Equivalently, we have to do this for the ensemble terms $\frac{1}{M}\sum_{i=1}^{M}y^\top(\tfrac{1}{D}\Phi_i\Phi_i^\top)^{-1}y$.

We first employ that we now that the operator norm of the difference $\tfrac{1}{MD}\Phi\Phi^\top - K$ is bounded by $\tfrac{\lambda_{min}}{2}$, implying that the eigenvalues of $\tfrac{1}{MD}\Phi\Phi^\top$ are bounded by $\tfrac{\lambda_{min}}{2}$ from below.

Thus, we have that $0 \le y^\top(\tfrac{1}{MD}\Phi\Phi^\top)^{-1}y \le \tfrac{2}{\lambda_{min}}\cdot\|y\|^2$ is bounded a.s. under the truncated distribution. Equivalently, this holds for the ensemble terms.

To bound the difference between the ensemble and single model, we can now employ a Hoeffding bound. This gives us that

$$\mathbb{P}\left[\left|\frac{1}{M}\sum_{i=1}^{M}y^\top(\tfrac{1}{D}\Phi_i\Phi_i^\top)^{-1}y - \mathbb{E}_{\tilde{\pi}_D}\left[y^\top(\tfrac{1}{D}\Phi_i\Phi_i^\top)^{-1}y\right]\right| \ge \epsilon\right] \le \exp\left(-M\epsilon^2\lambda_{min}\right)$$

and for the single model term:

$$\mathbb{P}\left[\left|y^\top(\tfrac{1}{MD}\Phi\Phi^\top)^{-1}y - \mathbb{E}_{\tilde{\pi}_{MD}}\left[y^\top(\tfrac{1}{MD}\Phi\Phi^\top)^{-1}y\right]\right| \ge \epsilon\right] \le \exp\left(-\epsilon^2\lambda_{min}\right)$$

Setting $\exp\left(-M\epsilon^2\lambda_{min}\right) + \exp\left(-\epsilon^2\lambda_{min}\right) \le \delta_3 := 2\cdot\exp\left(-\epsilon^2\lambda_{min}\right)$ and solving for $\epsilon$ gives us that:

$$\epsilon = \sqrt{\frac{1}{\lambda_{min}}\log\left(\frac{2}{\delta_3}\right)}$$

*Seventh step: Taking everything together.*

We can now employ a union bound to get that the probability that all three conditions are satisfied is at least $1 - \delta_1 - \delta_2 - \delta_3$ and thus we get the bound on the difference between the ensemble and single model in $\mathcal{L}_2$ norm.

Note that the fact that we worked under a truncated distribution in the previous step is not a problem, since the corresponding events have a lower probability under the the non-truncated distribution as long as we have already assumed the exclusion of all events where the difference between the empirical kernel matrix and the true kernel matrix is bigger than $\frac{\lambda_{min}}{2}$. □

### C.3. Variance of Ensemble Predictions

In the next step, we show the formula for the variance of a single model prediction under Gaussianity. Note that one could also get this result by slightly extending proofs by (Jacot et al., 2020).

**Lemma C.6** (Variance of single model predictions). *Under Gaussianity and assuming $D > N + 1$, the variance of single model prediction at a test point $x^*$ is given by*

$$\mathbb{V}_{\mathcal{W}}[h_{\mathcal{W}}^{(\text{LN})}(x^*)] = r_\perp^2 \frac{\|h_\infty^{(\text{LN})}\|_{\mathcal{H}}^2}{D - N - 1}, \tag{10}$$

*where $\|\cdot\|_{\mathcal{H}}$ is norm defined by the RKHS associated with kernel $k(\cdot, \cdot)$.*

*Proof.* We start by writing down the variance of the prediction of a single model:

$$\mathbb{V}_{\mathcal{W}}[h_{\mathcal{W}}^{(\text{LN})}(x^*)] = \mathbb{E}_{\mathcal{W}}[h_{\mathcal{W}}^{(\text{LN})}(x^*)^2] - \mathbb{E}_{\mathcal{W}}[h_{\mathcal{W}}^{(\text{LN})}(x^*)]^2$$

Using Theorem 3.2, the definition of the prediction of a single model and the definition of $W$ and $w_\perp$, we can expand this expression:

$$= \mathbb{E}_{\mathcal{W}}[\phi_{\mathcal{W}}^* \Phi_{\mathcal{W}}^\top (\Phi_{\mathcal{W}} \Phi_{\mathcal{W}}^\top)^{-1} yy^\top (\Phi_{\mathcal{W}} \Phi_{\mathcal{W}}^\top)^{-\top} \Phi_{\mathcal{W}} \phi_{\mathcal{W}}^{*\top}] - (h_\infty^{(\text{LN})}(x^*))^2$$

$$= \mathbb{E}_{W,w_\perp}[(r_\perp w_\perp^\top + c^\top W) W^\top R \left(R^\top W W^\top R\right)^{-1} yy^\top \left(R^\top W W^\top R\right)^{-\top} R^\top W (r_\perp w_\perp^\top + c^\top W)^\top]$$
$$- (h_\infty^{(\text{LN})}(x))^2$$

$$= \mathbb{E}_{W,w_\perp}[(r_\perp w_\perp^\top + c^\top W) W^\top \left(W W^\top\right)^{-1} R^{-\top} yy^\top R^{-1} \left(W W^\top\right)^{-\top} W (r_\perp w_\perp^\top + c^\top W)^\top]$$
$$- (h_\infty^{(\text{LN})}(x))^2$$

$$= (c^\top R^{-\top} y)^2 - (h_\infty^{(\text{LN})}(x))^2$$
$$+ 2 \cdot r_\perp^\top \mathbb{E}_{W,w_\perp}[w_\perp^\top W^\top (W W^\top)^{-1}] R^{-\top} yy^\top R^{-1} c$$
$$+ r_\perp^2 \mathbb{E}_{W,w_\perp}[w_\perp^\top W^\top (W W^\top)^{-1} R^{-\top} yy^\top R^{-1} (W W^\top)^{-T} W w_\perp]$$

Now we can see that the first two terms cancel out (since $h_\infty^{(\text{LN})}(x) = c^\top R^{-\top} y$) and the third term is zero by Lemma 3.1. We are left with the fourth term, which we can slightly rewrite:

$$\mathbb{V}_{\mathcal{W}}[h_{\mathcal{W}}^{(\text{LN})}(x^*)] = r_\perp^2 \mathbb{E}_{W,w_\perp}[w_\perp^\top W^\top (W W^\top)^{-1} R^{-\top} yy^\top R^{-1} (W W^\top)^{-T} W w_\perp]$$
$$= r_\perp^2 y^\top R^{-1} \mathbb{E}_{W,w_\perp}[(W W^\top)^{-T} W w_\perp w_\perp^\top W^\top (W W^\top)^{-1}] R^{-\top} y \tag{11}$$

Using the tower rule for conditional expectations, we have:

$$\mathbb{V}_{\mathcal{W}}[h_{\mathcal{W}}^{(\text{LN})}(x)] = r_\perp^2 y^\top R^{-1} \mathbb{E}_{W,w_\perp}[(W W^\top)^{-T} W w_\perp w_\perp^\top W^\top (W W^\top)^{-1}] R^{-\top} y$$
$$= r_\perp^2 y^\top R^{-1} \mathbb{E}_W[(W W^\top)^{-T} W \mathbb{E}_{w_\perp | W}[w_\perp w_\perp^\top | W] W^\top (W W^\top)^{-1}] R^{-\top} y$$

Since the Gaussianity assumption implies $W$ and $w_\perp$ are independent, we get:

$$\mathbb{V}_{\mathcal{W}}[h_{\mathcal{W}}^{(\text{LN})}(x)] = r_\perp^2 y^\top R^{-1} \mathbb{E}_W[(W W^\top)^{-T} W \mathbb{E}_{w_\perp}[w_\perp w_\perp^\top] W^\top (W W^\top)^{-1}] R^{-\top} y$$

Moreover, since by Gaussianity $w_\perp$ and $W$ are multivariate Gaussians with the identity matrix as covariance, we get (via the expected value of a Wishart and an inverse Wishart distribution; note that for getting this expected value, we need to assume that $D > N + 1$):

$$
\begin{aligned}
\mathbb{V}_{\mathcal{W}}[h_{\mathcal{W}}^{(\mathrm{LN})}(x)] &= r_\perp^2 y^\top R^{-1} \mathbb{E}_W[(WW^\top)^{-T}(WW^\top)(WW^\top)^{-1}] R^{-\top} y \\
&= r_\perp^2 y^\top R^{-1} \mathbb{E}_W[(WW^\top)^{-T}] R^{-\top} y \\
&= r_\perp^2 \frac{y^\top R^{-1} R^{-\top} y}{D - N - 1} \\
&= r_\perp^2 \frac{y^\top K^{-1} y}{D - N - 1}.
\end{aligned}
$$

Recognizing that $y^\top K^{-1} y = \|h_\infty^{(\mathrm{LN})}\|_{\mathcal{H}}^2$ (e.g. Wainwright, 2019, Ch. 12) completes the proof. $\qquad\square$

An equivalent argument does not work under the more general Assumption 2.1 since $w_\perp$ and $W$ are not necessarily independent. Even in the case of independence, $\mathbb{E}_W[(WW^\top)^{-1}]$ might not be known.

**Counterexample for subexponential case.** We now give an explicit counterexample showing that when only assuming uncorrelatedness between $W$ and $w_\perp$ the term

$$
E := \mathbb{E}_{W,w_\perp}[(WW^\top)^{-T} W w_\perp w_\perp^\top W^\top (WW^\top)^{-1}]
$$

from Eq. (11) depends on $x^*$ implying that the variance does not only depend on $x^*$ via $r_\perp^2$.

Let us assume $N = D = 1$ and let $W$ be uniformly distributed across the set $\left\{-\frac{4}{\sqrt{12.5}}, -\frac{3}{\sqrt{12.5}}, \frac{3}{\sqrt{12.5}}, \frac{4}{\sqrt{12.5}}\right\}$. Then we have $\mathbb{E}[W] = 0$ and $\mathbb{E}[W^2] = \frac{1}{2} \cdot \frac{16}{12.5} + \frac{1}{2} \cdot \frac{9}{12.5} = 1$.

Now consider an $x^*$ that produces a $w_\perp$ so that $w_\perp = \sqrt{2}$ when $W = \left\{-\frac{3}{\sqrt{12.5}}, \frac{3}{\sqrt{12.5}}\right\}$ and $w_\perp = 0$ otherwise. Then we have $\mathbb{E}[w_\perp^\top W] = 0$ and $\mathbb{E}[w_\perp^2] = 1$. The value of $E$ is now $\frac{12.5}{9}$.

Furthermore, consider an $x^*$ that produces a $w_\perp$ so that $w_\perp = \sqrt{2}$ when $W = \left\{-\frac{4}{\sqrt{12.5}}, \frac{4}{\sqrt{12.5}}\right\}$ and $w_\perp = 0$ otherwise. Then we have $\mathbb{E}[w_\perp^\top W] = 0$ and $\mathbb{E}[w_\perp^2] = 1$. The value of $E$ is now $\frac{12.5}{16}$.

# D. Proofs for Overparameterized Ridge Regression

## D.1. Difference between the Infinite Ensemble and Infinite Single Model

We begin with a lemma, which shows that the prediction of kernel regressors is Lipschitz-continuous in $\lambda$ for any $x^*$ and $\lambda \geq 0$. We will denote the kernel ridge regressor with regularization parameter $\lambda$ as $h_{\infty,\lambda}^{(RR)}$, as introduced in Sec. 3.4.

**Lemma D.1** (Bound on the difference between the kernel ridge regressors). *Let $\lambda, \lambda' \geq 0$ be two regularization parameters. Then, for any $x^* \in \mathcal{X}$ it holds that:*

$$
|h_{\infty,\lambda'}^{(RR)}(x^*) - h_{\infty,\lambda}^{(RR)}(x^*)| \leq \sqrt{n} \cdot C_1 \cdot |\lambda' - \lambda| \cdot \sqrt{y^T K^{-4} y}
$$

*where we assume $k(x_i, x^*) \leq C_1$ for all $i \in [N]$.*

*Proof.* We can write the kernel ridge regressors as $h_{\infty,\lambda}^{(RR)}(x^*) = \sum_{i=1}^n \alpha_{1,i} k(x_i, x^*)$ and $h_{\infty,\lambda'}^{(RR)}(x^*) = \sum_{i=1}^n \alpha_{2,i} k(x_i, x^*)$ with coefficients $\alpha_1$ and $\alpha_2$ given by:

$$
\begin{aligned}
\alpha_1 &= (K + \lambda I)^{-1} y \\
\alpha_2 &= (K + \lambda' I)^{-1} y
\end{aligned}
$$

We now write $y$ in the orthonormal basis of the eigenvectors of $K$, i.e. $y = \sum_{i=1}^n a_i v_i$. We call the corresponding eigenvalues of $K$ $d_1, \ldots, d_n > 0$.

The matrix $(K + \lambda I)^{-1}$ has the same eigenvectors as $K$ and the eigenvalues are $0 < \tilde{d}_i = \frac{1}{d_i + \lambda} \leq \frac{1}{\lambda}$. Thus, we can write $\alpha_1 = \sum_{i=1}^n a_i \frac{1}{d_i + \lambda} v_i$ and $\alpha_2 = \sum_{i=1}^n a_i \frac{1}{d_i + \lambda'} v_i$.

In the next step, we bound $\|\alpha_1 - \alpha_2\|_2^2$: Using the orthonormality of the eigenvectors, we get:

$$\|\alpha_1 - \alpha_2\|_2^2 = \sum_{i=1}^n \left( a_i \left( \frac{1}{d_i + \lambda} - \frac{1}{d_i + \lambda'} \right) \right)^2$$

Now we bound $\left| \frac{1}{\lambda + d_i} - \frac{1}{\lambda' + d_i} \right| \leq \left| \frac{\lambda' - \lambda}{\lambda \lambda' + (\lambda + \lambda')d_i + d_i^2} \right| \leq \frac{|\lambda' - \lambda|}{d_i^2}$ which gives us:

$$\|\alpha_1 - \alpha_2\|_2^2 \leq \sum_{i=1}^n \left( \frac{a_i |\lambda' - \lambda|}{d_i^2} \right)^2 \leq |\lambda' - \lambda|^2 y^T K^{-4} y$$

Using this result, we can bound the difference between the predictions of the two kernel regressors at a single point $x^*$:

$$|h_{\infty,\lambda}^{(RR)}(x^*) - h_{\infty,\lambda'}^{(RR)}(x^*)| = |\sum_{i=1}^n (\alpha_{1,i} - \alpha_{2,i}) k(x_i, x^*)| \leq \sum_{i=1}^n |\alpha_{1,i} - \alpha_{2,i}| k(x_i, x^*)$$

Since $k(x_i, x^*) \leq C_1$, we get (using the relation between the 1-norm and the 2-norm):

$$|f_\lambda(x^*) - f_{\lambda'}(x^*)| \leq C_1 \sum_{i=1}^n |\alpha_{1,i} - \alpha_{2,i}| \leq C_1 \|\alpha_1 - \alpha_2\|_2 \sqrt{n} \leq \sqrt{n} \cdot C_1 \cdot |\lambda' - \lambda| \cdot \sqrt{y^\top K^{-4} y}$$

$\square$

Using similar arguments, we now show that the expected prediction of RF regressors, i.e., the prediction of the infinite ensemble of RF regressors, is Lipschitz-continuous for any $x^*$ and $\lambda \geq 0$:

**Lemma D.2** (Bound on the difference between expected RF Regressors). *Under Assumption 2.1 and Assumption 3.4, the expected value of the prediction of RF regressors is Lipschitz-continuous in $\lambda$ for any $x^*$ and $\lambda \geq 0$, i.e., for any $x^*$ it holds that:*

$$|\bar{h}_{\infty,\lambda'}^{(RR)}(x^*) - \bar{h}_{\infty,\lambda}^{(RR)}(x^*)| \leq \|c^\top R^{-\top}\| \|y\| D C_2 |\lambda' - \lambda|$$

*where $C_2$ is a constant depending on the distribution of $\Phi$.*

*Proof.* We use the characterization of $\bar{h}_{\infty,\lambda}^{(RR)}(x^*)$ from Eq. (9), which gives us the difference as

$$\left| c^\top \mathbb{E}_{W,w_\perp} \left[ WW^\top \left( \left( WW^\top + D \cdot \lambda' \cdot R^{-\top} R^{-1} \right)^{-1} - \left( WW^\top + D \cdot \lambda' \cdot R^{-\top} R^{-1} \right)^{-1} \right) \right] R^{-\top} y \right|.$$

We can now reverse some steps we made to get this characterization and write it in terms of $\Phi$ again:

$$\left| c^\top R^{-\top} \mathbb{E}_{\mathcal{W}} \left[ \Phi_{\mathcal{W}} \Phi_{\mathcal{W}}^\top \left( \left( \Phi_{\mathcal{W}} \Phi_{\mathcal{W}}^\top + D \cdot \lambda' \cdot I \right)^{-1} - \left( \Phi_{\mathcal{W}} \Phi_{\mathcal{W}}^\top + D \cdot \lambda \cdot I \right)^{-1} \right) \right] y \right|.$$

And now, using Jensen's inequality and the convexity of the two-norm, we can pull out the expected value to the outside of the difference:

$$\|c^\top R^{-\top}\| \cdot \mathbb{E}_{\mathcal{W}} \left[ \|\Phi_{\mathcal{W}} \Phi_{\mathcal{W}}^\top \left( \left( \Phi_{\mathcal{W}} \Phi_{\mathcal{W}}^\top + D \cdot \lambda' \cdot I \right)^{-1} - \left( \Phi_{\mathcal{W}} \Phi_{\mathcal{W}}^\top + D \cdot \lambda \cdot I \right)^{-1} \right) y \| \right].$$

Similarly to the proof of Lemma D.1, we can write $y$ in the orthonormal basis of the eigenvectors of $\Phi \Phi^\top$ (note that we drop the subscript $\mathcal{W}$ for notational simplicity), i.e. $y = \sum_{i=1}^n a_i v_i$. Furthermore we define the eigenvalues of $\Phi \Phi^\top$ as $d_1, \ldots, d_n > 0$. The matrix $(\Phi \Phi^\top + D \cdot \lambda I)^{-1}$ again has the same eigenvectors as $\Phi \Phi^\top$ and the eigenvalues are $0 < \frac{1}{d_i + D \cdot \lambda} \leq \frac{1}{D \cdot \lambda}$.

Multiplying $y$ with $\Phi \Phi^\top (\Phi \Phi^\top + D \cdot \lambda I)^{-1}$ and $\Phi \Phi^\top (\Phi \Phi^\top + D \cdot \lambda' I)^{-1}$ then gives us:

$$\Phi \Phi^\top (\Phi \Phi^\top + D \cdot \lambda I)^{-1} y = \sum_{i=1}^n a_i \frac{d_i}{d_i + D \cdot \lambda} v_i$$
$$\Phi \Phi^\top (\Phi \Phi^\top + D \cdot \lambda' I)^{-1} y = \sum_{i=1}^n a_i \frac{d_i}{d_i + D \cdot \lambda'} v_i$$

We can now calculate the difference of these two vectors using the orthonormality of the eigenvectors:

$$\|\Phi\Phi^\top(\Phi\Phi^\top + D \cdot \lambda' I)^{-1}y - \Phi\Phi^\top(\Phi\Phi^\top + D \cdot \lambda I)^{-1}y\|_2^2 = \sum_{i=1}^n \left(a_i\left(\frac{d_i}{d_i + D\cdot\lambda} - \frac{d_i}{d_i + D\cdot\lambda'}\right)\right)^2$$

Now we look at the difference between the two coefficients and see that for each $i$, we have:

$$\left|\frac{d_i}{d_i + D\cdot\lambda} - \frac{d_i}{d_i + D\cdot\lambda'}\right| \leq \frac{D\cdot|\lambda'-\lambda|}{d_i}$$

Thus, we have that the difference is bounded by:

$$\|\Phi\Phi^\top(\Phi\Phi^\top + D\cdot\lambda' I)^{-1}y - \Phi\Phi^\top(\Phi\Phi^\top + D\cdot\lambda I)^{-1}y\|_2^2 \leq \frac{D^2\cdot|\lambda-\lambda'|^2}{d_N^2}\|y\|_2^2.$$

All together, we can now bound the difference of the expected values of the predictions of RF regressors via:

$$|\bar{h}_{\infty,\lambda'}^{(RR)}(x^*) - \bar{h}_{\infty,\lambda}^{(RR)}(x^*)| \leq \|c^\top R^{-\top}\|\|y\|D|\lambda' - \lambda|\mathbb{E}_{d_N}\left[\frac{1}{d_N}\right]$$

Since $\operatorname{tr}((\Phi\Phi^\top)^{-1}) = \sum_{i=1}^n \frac{1}{d_i}$, and the trace is a linear operator, we can write:

$$\mathbb{E}_{d_N}\left[\frac{1}{d_N}\right] \leq \mathbb{E}_{\mathcal{W}}\left[(\operatorname{tr}(\Phi_{\mathcal{W}}\Phi_{\mathcal{W}}^\top)^{-1})\right] = \operatorname{tr}(\mathbb{E}_{\mathcal{W}}\left[(\Phi_{\mathcal{W}}\Phi_{\mathcal{W}}^\top)^{-1}\right]) =: C_2$$

which is finite whenever $\mathbb{E}_{\mathcal{W}}\left[(\Phi_{\mathcal{W}}\Phi_{\mathcal{W}}^\top)^{-1}\right]$ is finite, i.e. Assumption 3.4 holds. $\qquad\square$

Using Lemma D.1 and Lemma D.2 we can now show that the difference between the infinite ensemble where each model has ridge $\lambda$ and the infinite single model with ridge $\lambda$ is Lipschtiz-continuous in $\lambda$ for $\lambda \geq 0$:

*Theorem 3.5 (Restated).* Under Assumptions 2.1 and 3.4, the difference $|\bar{h}_{\infty,\lambda}^{(RR)}(x^*) - h_{\infty,\lambda}^{(RR)}(x^*)|$ between the infinite ensemble and the single infinite-width model trained with ridge $\lambda \geq 0$ is Lipschitz-continuous in $\lambda$. The Lipschitz constant is independent of $x^*$ for compact $\mathcal{X}$.

*Proof.* We bound difference $\left||\bar{h}_{\infty,\lambda'}^{(RR)}(x^*) - h_{\infty,\lambda'}^{(RR)}(x^*)| - |\bar{h}_{\infty,\lambda}^{(RR)}(x^*) - h_{\infty,\lambda}^{(RR)}(x^*)|\right|$ by using first the inverse, then the normal triangle inequality:

$$\left||\bar{h}_{\infty,\lambda'}^{(RR)}(x^*) - h_{\infty,\lambda'}^{(RR)}(x^*)| - |\bar{h}_{\infty,\lambda}^{(RR)}(x^*) - h_{\infty,\lambda}^{(RR)}(x^*)|\right|$$
$$\leq |\bar{h}_{\infty,\lambda'}^{(RR)}(x^*) - \bar{h}_{\infty,\lambda}^{(RR)}(x^*) + h_{\infty,\lambda}^{(RR)}(x^*) - h_{\infty,\lambda'}^{(RR)}(x^*)|$$
$$\leq |\bar{h}_{\infty,\lambda'}^{(RR)}(x^*) - \bar{h}_{\infty,\lambda}^{(RR)}(x^*)| + |h_{\infty,\lambda}^{(RR)}(x^*) - h_{\infty,\lambda'}^{(RR)}(x^*)|$$

Using the bound from Lemma D.1 and Lemma D.2 (and summarizing the the corresponding constants as $c_1$ and $c_2$) we can bound this by:

$$|\bar{h}_{\infty,\lambda'}^{(RR)}(x^*) - h_{\infty,\lambda'}^{(RR)}(x^*)| - |\bar{h}_{\infty,\lambda}^{(RR)}(x^*) - h_{\infty,\lambda}^{(RR)}(x^*)| \leq c_1|\lambda' - \lambda| + c_2|\lambda' - \lambda|$$

Thus we have Lipschitz-continuity in $\lambda$ for $\lambda \geq 0$.

The Lipschitz constant is independent of $x^*$ for $\mathcal{X}$ compact since the Lipschitz constants from Lemma D.1 and Lemma D.2 depend on $x^*$ in a continuous fashion. $\qquad\square$

Note that an equivalent argument in combination with (Jacot et al., 2020)[Proposition 4.2], i.e. $\tilde{\lambda} \leq \frac{\gamma}{\gamma-1}\lambda$, directly gives the Lipschitz-continuity in $\lambda$ for $\lambda \geq 0$ for the difference between the infinite ensemble and the infinite-width single model with effective ridge in the overparameterized regime.

# E. Underparameterized Ensembles

Here, we offer a proof that infinite, unregularized, underparameterized RF ensembles are equivalent to kernel ridge regression under a transformed kernel function. We emphasize the difference from the overparameterized case—the central focus of our paper—in which the infinite ensemble is equivalent to a ridgeless kernel regressor. Thus, underparameterized ensembles induce regularization, while overparameterized ensembles do not.

Other works have explored the ridge behavior of underparameterized RF ensembles (Kabán, 2014; Thanei et al., 2017; Bach, 2024b); however, these works often focus on an equivalence in generalization error whereas we establish a pointwise equivalence. To the best of our knowledge, the following result is novel:

**Lemma E.1.** *If the expected orthogonal projection matrix* $\mathbb{E}_{\tilde{W}}\left[R^\top \tilde{W}\left(W^\top RR^\top W\right)^{-1}\tilde{W}^\top R\right]$ *is well defined, and a contraction (i.e., singular values strictly less than 1), then the infinite underparameterized RF ensemble* $\bar{h}_\infty^{(LN)}(x^*)$ *is equivalent to kernel ridge regression under some kernel function* $\tilde{k}(\cdot,\cdot)$.

*Proof.* When $D < N$, the infinite ridgeless RF ensemble is given by

$$\bar{h}_\infty^{(LN)}(x^*) = \mathbb{E}_\mathcal{W}\left[\frac{1}{D}\sum_{j=1}^D \phi(\omega_j, x^*)\left(\frac{1}{D}\Phi_\mathcal{W}^\top \Phi_\mathcal{W}\right)^{-1}\Phi_\mathcal{W}^\top\right]y$$
$$= \mathbb{E}_{W,w_\perp}\left[\left(r_\perp w_\perp^\top + c^\top W\right)\left(W^\top RR^\top W\right)^{-1}W^\top\right]Ry, \tag{12}$$

where $W, w_\perp, r_\perp, c, R$ are as defined in Sec. 2. Defining the following block matrices:

$$\tilde{W} = \begin{bmatrix} W \\ w_\perp^\top \end{bmatrix} \in \mathbb{R}^{(N+1)\times D}, \qquad \tilde{R} = \begin{bmatrix} R \\ 0 \end{bmatrix} \in \mathbb{R}^{(N+1)\times N}, \qquad \tilde{c} = \begin{bmatrix} c \\ r_\perp \end{bmatrix} \in \mathbb{R}^{(N+1)},$$

we can rewrite Eq. (12) as

$$\bar{h}_\infty^{(LN)}(x^*) = \tilde{c}^\top\left(\mathbb{E}_{\tilde{W}}\left[\tilde{W}\left(W^\top RR^\top W\right)^{-1}\tilde{W}^\top\right]\right)\tilde{R}y.$$

By adding and subtracting $\tilde{R}\tilde{R}^\top$ inside the outer parenthesis, we can massage this expression into kernel ridge regression in a transformed coordinate system:

$$\bar{h}_\infty^{(LN)}(x^*) = \tilde{c}^\top\left(\tilde{R}\tilde{R}^\top + \underbrace{\left(\mathbb{E}_{\tilde{W}}\left[\tilde{W}\left(W^\top RR^\top W\right)^{-1}\tilde{W}^\top\right]\right)^{-1} - \tilde{R}\tilde{R}^\top}_{:=\tilde{A}}\right)^{-1}\tilde{R}y.$$
$$= \tilde{c}^\top \tilde{A}^{-1}\tilde{R}\left(\tilde{R}^\top \tilde{A}^{-1}\tilde{R} + I\right)^{-1}y. \tag{13}$$

Applying the Woodbury inversion lemma to $\tilde{A}^{-1}$, we have:

$$\tilde{A}^{-1} = \mathbb{E}_{\tilde{W}}\left[\tilde{W}\left(W^\top RR^\top W\right)^{-1}\tilde{W}^\top\right]$$
$$+ \mathbb{E}_{\tilde{W}}\left[\tilde{W}\left(W^\top RR^\top W\right)^{-1}W^\top R\right]\left(I - \mathbb{E}_W[P_W]\right)^{-1}\mathbb{E}_{\tilde{W}}\left[R^\top W\left(W^\top RR^\top W\right)^{-1}\tilde{W}^\top\right], \tag{14}$$

where $P_W$ is the (random) orthogonal projection matrix onto the span of the columns of $R^\top W$:

$$P_W = R^\top W\left(W^\top RR^\top W\right)^{-1}W^\top R.$$

Because $P_W$ is an orthogonal projection matrix, we have that $\|P_W\|_2 = 1$, and thus (by Jensen's inequality) $\|\mathbb{E}_W[P_W]\|_2 \le 1$. If this inequality is strict so that $I - \mathbb{E}_W[P_W]$ is invertible, we have by inspection of Eq. (14) that $\tilde{A}$ is positive definite. Therefore, the block matrix

$$\begin{bmatrix} \tilde{R}^\top \\ \tilde{c}^\top \end{bmatrix}\tilde{A}^{-1}\begin{bmatrix} \tilde{R} & \tilde{c} \end{bmatrix} = \begin{bmatrix} \tilde{R}^\top \tilde{A}^{-1}\tilde{R} & \tilde{R}^\top \tilde{A}^{-1}\tilde{c} \\ \tilde{c}^\top \tilde{A}^{-1}\tilde{R} & \tilde{c}^\top \tilde{A}^{-1}\tilde{c} \end{bmatrix} \tag{15}$$

is also positive definite and thus the realization of some kernel function $\tilde{k}(\cdot, \cdot)$; i.e.

$$
\begin{bmatrix} \tilde{R}^\top \tilde{A}^{-1} \tilde{R} & \tilde{R}^\top \tilde{A}^{-1} \tilde{c} \\ \tilde{c}^\top \tilde{A}^{-1} \tilde{R} & \tilde{c}^\top \tilde{A}^{-1} \tilde{c} \end{bmatrix} = \begin{bmatrix} \tilde{k}(x_1, x_1) & \cdots & \tilde{k}(x_1, x_N) & \tilde{k}(x_1, x^*) \\ \vdots & \ddots & \vdots & \vdots \\ \tilde{k}(x_N, x_1) & \cdots & \tilde{k}(x_N, x_N) & \tilde{k}(x_N, x^*) \\ \tilde{k}(x^*, x_1) & \cdots & \tilde{k}(x^*, x_N) & \tilde{k}(x^*, x^*) \end{bmatrix}.
$$

Note that if $\tilde{A} = I$ then by Eq. (1) we recover the original kernel matrix

$$
\begin{bmatrix} \tilde{R}^\top \tilde{R} & \tilde{R}^\top \tilde{c} \\ \tilde{c}^\top \tilde{R} & \tilde{c}^\top \tilde{c} \end{bmatrix} = \begin{bmatrix} k(x_1, x_1) & \cdots & k(x_1, x_N) & k(x_1, x^*) \\ \vdots & \ddots & \vdots & \vdots \\ k(x_N, x_1) & \cdots & k(x_N, x_N) & k(x_N, x^*) \\ k(x^*, x_1) & \cdots & k(x^*, x_N) & k(x^*, x^*) \end{bmatrix}.
$$

Thus, the underparameterized ensemble in Eq. (13) simplifies to

$$
\bar{h}_\infty^{(LN)}(x^*) = \begin{bmatrix} \tilde{k}(x^*, x_1) & \cdots & \tilde{k}(x^*, x_N) \end{bmatrix} \left( \begin{bmatrix} \tilde{k}(x_1, x_1) & \cdots & \tilde{k}(x_1, x_N) \\ \vdots & \ddots & \vdots \\ \tilde{k}(x_N, x_1) & \cdots & \tilde{k}(x_N, x_N) \end{bmatrix} + I \right)^{-1} y,
$$

which is kernel ridge regression with respect to the kernel $\tilde{k}(\cdot, \cdot)$. $\qquad\square$

