# OpenReview forum: "Theoretical Limitations of Ensembles in the Age of Overparameterization"
_ICML.cc/2025/Conference — ICML 2025 oral_

### Official Review · Reviewer_e9Q9 · 2025-03-07

**Overall Recommendation:** 5

**Summary:**

This paper studies ensembles of M random feature networks when the number of features D is greater than the number of data points N (overparameterized regime). Large ensembles are found to be asymptotically equivalent to a single large network and convergence bounds are given for finite M. There are numerical experiments that illustrate the key results and implications for real networks are discussed.

**Claims And Evidence:**

The main "claim" is that large ensembles become equivalent to large-width models. Besides the theory (reviewed below), numerical experiments were run with random features as well as neural networks (Figs 2, 3). Overall, I found the claims fairly clear and well-supported. There seemed to be some exaggeration of how well the numerical results fit with the theory. For instance:
* Fig 2 right is said to show a "hockey stick" pattern, however this isn't nearly as obvious on the right (neural networks) versus at left (with RF ensemble).
* Fig 3 (left and right) both show ensembles outperforming single models despite the main "result" being an argument that these are equivalent as the networks/ensembles get larger. In fact, the gap at left seems to be growing as the network grows, which isn't inconsistent with the theory and in fact the ensemble outperforms the kernel, its infinite limit. I think the issue here is the interpretation of the theory; the theory tells about the infinite limit but isn't very relevant in the finite case where ensemble will have a variance-limiting effect. In this plot it would help to show both means and shaded regions for variance across instantiations of single models/ensembles. My guess is the variability of the curve at right is due to variance across single networks. Can you explain this discrepancy?
* Fig 4: I found this convincing; it might be better if plotted on the same axes.

**Essential References Not Discussed:**

Early work by Radford Neal (1996) that was the first to connect random feature networks with kernels. I usually cite that and CKI Williams (1997) when writing papers in this area.

**Experimental Designs Or Analyses:**

I read the supplemental description of the experimental setup and didn't have any issues with that.

**Methods And Evaluation Criteria:**

The datasets and methods seem fine for a theoretical study. There wasn't much discussion of the effect of input dimension on the results. I don't think this affects things, but is there a high-dimensional regime when the input dimension is large where these results break down?

**Other Comments Or Suggestions:**

Typos/small points/clarification needed:
* Eqn (1): Notation $[ . ]_j$ should be explained since it seems to refer to both column Nx1 and 1xN row vectors
* pg 3: Similar to above, MATLAB-like [W; w] notation should be described.
* pg 3: $\phi^*$ isn't defined
* pg 4 line 178: order of "ridge(less)" is reversed from least-norm and RR expressions that come later; I suggest reversing their order for clarity.
* Assumption 2.1: The expression $w_i w_{\perp i}$ is unclear, since $w_i$ is a vector and the other term is a scalar. Do you really mean their product or do you mean a column vector that comes from concatenating them? I suspect a typo here.
* Line 250 right column: "RF ensembles are equivalent to the ridgeless..." I think you can, at best, say they are "close to" here given the evidence you present.
* Lemma C.5: $z_i$ seems like it should be $x_i$.
* Inconsistent notation "Var" and $\mathbb{V}$ used for variance
* Figure 5 doesn't seem to be referred to in the paper itself
* Assumption 3.4: Can you clarify what you mean by finiteness of the matrix here? It would require the matrix to be invertible but also finite in expectation?

**Other Strengths And Weaknesses:**

I found the paper overall clear and the results interesting. My main issues are with some of the clarity in presentation.

**Questions For Authors:**

* Can you explain the discrepancy between the curves in Fig 2? This should be discussed in the paper
* I am somewhat familiar with the results in the paper by Ruben et al (2024) that also study random feature ensembles using a different theory. These related results aren't discussed in much detail, although the conclusion "NO FREE LUNCH FROM RANDOM FEATURE ENSEMBLES" is close to those of the current paper. Can you discuss?
* Can you discuss in more detail, perhaps at the end of section 3.3, when we expect the RF ensemble variance to capture Bayesian uncertainty or not? Does it ever work?
* In Sec 3.4 you make the point about ensembles with the same $\lambda$ converging. There are some results out there that show network width acts as an effective regularization (RF models at finite width are close to the kernel predictor with a modified ridge parameter, c.f. Bordelon, Canatar, Pehlevan, 2020). Wouldn't it then be best to use a different ridge parameter to compare ensembles with width $D$ to a single network of width $MD$?

**Relation To Broader Scientific Literature:**

I am not sure if the current results are super "novel" in that I think some of these results are known (e.g. Ruben et al, 2024). The idea that the variance across random features isn't indicative of a Bayesian measure of uncertainty, to me, wasn't surprising since I've always thought of this variance as being due to ensemble randomness rather than randomness in the training data. For instance, the theory behind using orthogonal random features or the FastFood random feature method was mostly concerned with keeping this variance low.

I don't think this means the current work isn't important. I think the generality of the assumptions taken here make these results nicely applicable in settings that haven't been considered before.

The discussion of the paper mentions the work of Abe et al. (2022) as well as in the introduction, but to understand what "recent empirical findings" the current paper is supposed to reproduce, it would be helpful for the reader to get a quick overview of those.

**Theoretical Claims:**

I read the main paper and found the results well-explained and intuitive. However, I did not check the proof supplement in detail.
I have some comments on the clarity of the mathematical presentation that I will list below.

---

> ### Author Rebuttal · Authors · 2025-03-31
>
> We thank the reviewer for their detailed review, as well as their suggestions for improving our paper. Below, we address the concerns and questions raised, and outline the changes we will make in response.
>
> **Claims And Evidence:**
>
> - **Clarification of "hockey stick" pattern in Fig. 2:** We agree and will adjust our wording in the revised manuscript to reflect that the "hockey stick" pattern is less pronounced for neural networks, likely due to effects that cannot be captured from an RF analysis.
> - **Interpretation of performance gap and variability in Fig. 3:** Thank you for highlighting this point. Indeed, the variability of the curve on the right arises due to variance across single model instantiations. We will update Fig. 3 accordingly, adding shaded regions to represent this variance. Regarding why the gap at the left grows, this is likely due to a) the discrepancy between the finite and infinite cases, and b) numerical instabilities when working with ReLU models (for a more precise explanation, see Appx. A.2). As an example of an experiment where this does not happen, compare Fig. 12 (right), which we deliberately did not include in the paper since this uses the Gaussian Error function.
> - **Plotting suggestion for Fig. 4:** Thank you for this suggestion – we will update Fig. 4 to use shared axes.
>
> **Relation to Prior Work and Additional References:**
>
> We will add a brief summary of Abe et al. (2022) and cite Neal (1996) and Williams (1997).
>
> **Other Comments or Suggestions:**
>
> We appreciate your edits and suggestions for clarity, which we will incorporate into our revised version. We address a few specific points below.
>
> - **Clarification of Assumption 2.1:** We indeed intended the product of the vector and scalar (which produces a vector again). Following standard extensions of subexponentiality to vector valued random variables, our theory relies on any linear combination of the entries of $w\_i w\_{\\bot i}$ being subexponential. We will clarify this point in the revision.
> - **Phrasing on line 250, right column:** We agree and will revise this phrasing accordingly.
> - **Clarification of Assumption 3.4:** We require that the entries of the expected value of the inverse matrix are finite, i.e., that the corresponding expected value exists. This obviously also requires that, for almost all instantiations of the matrix $\\Phi\_{\\mathcal{W}}$, the matrix $\\Phi\_{\\mathcal{W}} \\Phi\_{\\mathcal{W}}^\\top$ is invertible.
>
> **Questions for Authors:**
>
> - **Relation to Ruben et al. (2024):** We agree that Ruben et al. (2024) should be discussed in more detail and will include this in the revision; thank you. We would also note that this work is concurrent to ours and should be read as such. Both works conclude that ensembles of overparameterized random feature models do not outperform a single larger model with an equivalent total feature budget. However, we explicitly focus on the overparameterized and zero/small ridge regime, while Ruben et al. primarily analyze generalization under optimal ridge parameters in both the over- and underparameterized regimes, with only brief consideration of our regime. Additionally, our analysis also explicitly considers uncertainty quantification, and we do not rely on Gaussianity assumptions, contrasting their Gaussian universality assumption.
> - **Ensemble variance and Bayesian uncertainty:** As briefly mentioned in Sec. 3.3, only "with Gaussian features, ensemble variance admits a Bayesian interpretation," by which we mean that the ensemble variance matches the posterior variance of a GP with the same limiting kernel. We will ensure this is clearer in the revised version.
> - **Different ridge parameters in Sec. 3.4:** We appreciate this suggestion and agree it would be interesting to explore. Note, however, that when investigating the finite-width regime, we assumed $\\lambda \= 0$, while in Sec. 3.4, we specifically discuss the infinite-width limit. Additionally, our current focus was explicitly on the small ridge regime, where any implicit regularization parameters (such as $\\tilde{\\lambda}$ from Jacot et al. (2020)) are expected to be very small; therefore, variations in this parameter are likely to have only a minor impact in our setting.
>
> We hope these responses and adjustments clarify our contributions and address your feedback. If you have further questions or suggestions, we would be happy to address them.

---

> > ### Comment · Reviewer_e9Q9 · 2025-04-02
> >
> > Thanks for your responses.
> > Now reading my comment about the vector times scalar issue seems silly, of course it should be interpreted that way.
> >
> > I've revised my score to a 5

---

### Official Review · Reviewer_hF3L · 2025-03-10

**Overall Recommendation:** 4

**Summary:**

This paper presents a theoretical analysis of ensembles of overparametrized models (more parameters than training data) in which authors claim that an Infinite Ensemble is equivalent to an Infinite-Width Single Model. This analysis is done by using the equivalence between random feature regressors (RF) and neural networks. In particular, the authors aim at answering two key questions:

- Do ensembles of overparameterized models provide generalization or robustness benefits over a single (very large) model trained on the same data?
- What does the predictive variance of overparameterized ensembles measure?

In order to tackle these questions, the authors compares the Infinite Ensembles and the Infinite-Width Single Models for the RF model. In simple terms, the model resulting from aggregating (i.e., computing the mean over the distribution of $\omega$) infinitely many models is pointwise equivalent to the infinite-width model. This result is based on Assumption 2.1 asking $\Phi$ to be full rank and controlling the dependence with W and $\phi(w_i,x*)$. It is important to underline that Theorem 3.2 is independent of the RF distribution. In order to complete the comparison and analysis on ensemble methods, in Section 3.2 authors assume limited computational budget. Here, in Theorem 3.3 they bound the distance between M models with D features and one single of MD random features. Figure 3, exhibits the theoretical finding, underlying the small difference in the generalization error. Since an important feature of ensemble methods is their ability to give measures of uncertainty, authors study in section 3.3. how the predictive variance of the two compared approaches behaves, giving details of Gaussian and general features. They conclude their analysis in section 3.4 accounting for the role of $\lambda$ (the ridge parameter in the loss function) for the infinite ensemble and infinite-width model. As it was predictable, allowing $\lambda$ to get higher values, affects the distance, since it is a parameter controlling for parametrization.

## Update after rebuttal
I thank the authors for integrating some comments in order to better clarify the work. I appreciate your efforts in answering my doubts and comments, which are convincing and clear. I therefore confirm to accept the paper.

**Claims And Evidence:**

All claims are well supported and conclusions are realistic. This is a mainly theoretical paper in which claims are always supported by some kind of experiment.

**Essential References Not Discussed:**

Even though they cite [1] I think they missed the opportunity to introduce the term joint (or end-to-end) training, which gives a good vision of why ensembles can be similar to single models depending on how we train them.
I think they missed [2] and [3], which showed similar results with less theoretical derivations and explained consequences of joint training, respectively.

[1] Jeffares, A., Liu, T., Crabbé, J., & van der Schaar, M. (2023). Joint training of deep ensembles fails due to learner collusion. Advances in Neural Information Processing Systems, 36, 13559-13589.

[2] Webb, A., Reynolds, C., Chen, W., Reeve, H., Iliescu, D., Lujan, M., & Brown, G. (2021). To ensemble or not ensemble: When does end-to-end training fail?. In Machine Learning and Knowledge Discovery in Databases: European Conference, ECML PKDD 2020, Ghent, Belgium, September 14–18, 2020, Proceedings, Part III (pp. 109-123). Springer International Publishing.

[3] Abe, T., Buchanan, E. K., Pleiss, G., & Cunningham, J. P. (2022). The best deep ensembles sacrifice predictive diversity. In I Can't Believe It's Not Better Workshop: Understanding Deep Learning Through Empirical Falsification.

**Experimental Designs Or Analyses:**

I think they could include more cases a part from the synthetic and California Housing. Specially, I think the experimental setup should be explicitly mentioned in the main text.
Despite that, this paper is mainly theoretical and I think the experiments support well their findings.

**Methods And Evaluation Criteria:**

Since the paper is mainly theoretical, the experiments carried are not very extensive. Only synthetic data and a single real world dataset (California Housing) are used, what limits a bit the validity of the evidences. However, all experimental result are suitable for the claim aiming to support and provide a good proof of the theoretical results.
Authors claim that this method is independent of the $\omega$ distribution but only use synthetic data from a Normal one and one single real world dataset.
The idea of using RF regressors as Neural Networks, even not novel, is correctly chosen.

**Other Comments Or Suggestions:**

C1- The whole paper is well written and no substantial error was found.

C2- In line 297 the notation for the L_2 norm is quite unusual and $\_2$ would fit better.

C3- It is not clear the connection between Figure 5 and Theorem 3.5 as there is no reference in text to Figure 5 and apparently one is dealing with $\bar{h}^{LS}\_{\infty}(x^*)$ and the other $\bar{h}^{RR}\_{\infty,\lambda}(x^*)$. Even though it is mentioned the reasoning in Appendix 4, I suggest a more detailed and clear explanation as the evolution w.r.t $\bar{h}^{RR}\_{\infty,\lambda}(x^*)$ is never shown.

C4- In Figure 1, authors write “We note no perceptible difference between the two.” which is quite vague. It would be more appropriate to quantify the distance. Also would it good to actually see the evolution of the graph, when increasing M, instead of just two values?

C5- Assumptions are more or less explained, even though an intuitive idea of them would help the reader in understanding the main results. In the text, there is no mention of how feasible are these assumptions.

C6- I understand that when authors say in line 295 (2nd column) that ensemble provide no additional robustness benefits they talk from a variance point of view when the number of parameters is fixed. However, in the right plot of Figure 3 we can see that ensembles show more stability to the total number of parameters. Is this only a consequence of averaging the models or has other explanations?

C7- I know the length restriction of the paper is limiting, but counterexamples like Figure 8 in Appendix A.2. and an explanation to them would reinforce the findings.

C8- In the experimental section, authors explain to take parameters $\omega$ from the standard normal distribution. This is a bit contrasting with their claim of generality of their results with respect to the parameter distribution. Also, part of the novelty of this paper is exactly the generality of results under a certain distribution $\pi(\cdot)$, hence it would have been better to provide a small example with another distribution.

C9- In Appendix A.2. authors claim that $\Phi_{\Omega}\Phi^T_{\Omega}$ is not almost surely invertible when using the ReLU activation function, hence contrasting assumption 2 in 2.1. I understand the approach to tackle this issue, but what about considering another activation function? Would it be interesting to analyze the impact of several activation functions on this assumption?

C10- Line 80 repeats “model” twice.

C11- Have you considered the effect of overparametrization in the double descent phenomena?

**Other Strengths And Weaknesses:**

The paper is quite technical and I admit I am not an expert in the subject of kernel functions. However, the mathematical details convinced me and results are coherent with their goals. Moreover, they approached the comparison of the single model and the aggregated model in many ways, resulting in a complete and detailed analysis. The conclusions correctly reflects their finding suggesting that the benefits of overparametrized model may be explained by their similarity to larger models.
I find the conclusion section very well written.

**Questions For Authors:**

Q1- The comparison between Deep ensembles and single large models assumes that the training process is the same, right? Models in Deep ensembles are trained independently because one seeks diversity among models. It is true that a set of models could be trained end-to-end since joint function is fully-differentiable, but as [1] demonstrates this is, effectively, equivalent to a single wide model.

Q2-  A part from the predictive variance, how could the single model decouple epistemic and aleatoric uncertainty?

Q3- How feasible are the assumptions 2.1 necessary to prove the main Results? Since the matrix $\Phi_{\Omega}\Phi^T_{\Omega}$ is not invertible (hence non-positive definite) in the case of ReLU, it is reasonable to consider other activation functions? What would be their impact on the results? Would the assumption be satisfied?

Q4- Besides the technical proof, how can you justify the statement “In practical terms, this result indicates that for sufficiently small values of λ, the predictions of large ensembles and large single models remain nearly indistinguishable” regarding the impact of the the regulariser parameter?

**Relation To Broader Scientific Literature:**

According to comparing ensembles with single models, the paper presents good literature. However, literature related to uncertainty quantification is a bit scarce and they do not consider the fact that a single model cannot distinguish between epistemic and aleatoric uncertainty.

**Theoretical Claims:**

All theoretical claims seem to be well suited and properly derived. Appendix shows all demonstrations. The important claims have been checked and properly understood.

---

> ### Author Rebuttal · Authors · 2025-03-31
>
> We thank the reviewer for their valuable comments and feedback on our paper. Below, we address the concerns and questions raised and outline the changes we plan to make to the manuscript based on your suggestions.
>
> **Regarding the Experiments:**
>
> We agree that explicitly including the experimental setup in the main text will help clarity. Thus, we will move these details from the appendix into the main body of the paper.
>
> **Additional literature:**
>
> Thanks for pointing out the additional papers. We will discuss \[1\], \[2\] and \[3\] in our related works section. Furthermore, we will add relevant references in the field of uncertainty quantification to our related works section and we will address the distinction between epistemic and aleatoric uncertainty.
>
> **Specific Comments:**
>
> * **C2 (Line 297, L2-norm notation):** We will correct the notation as suggested.
> * **C3 (Connection between Figure 5 and Theorem 3.5):** We will add a short explanation of and a reference to Figure 5 in the main text.
> * **C4 (Quantifying the difference in Figure 1):** We will include a more detailed plot showing the evolution with increasing ensemble size $M$ in the appendix and reference this in the figure description.
> * **C5 (Feasibility and intuitiveness of assumptions):** We will add a brief intuitive explanation of Assumption 2.1, noting that these assumptions closely hold in most relevant scenarios including e.g., for distributions with bounded support.
> * **C6 (Stability vs. robustness in Figure 3):** Indeed, the observed stability in Figure 3 arises from averaging effects. As suggested by reviewer e9Q9 we will add shaded regions for variance across instantiations of single models/ensembles
> * **C7 (Counterexamples in Appendix A.2):** Due to length constraints, we cannot incorporate Figure 8 directly into the main text but will ensure it is more prominently referenced in the main text.
> * **C8 (Experiments beyond standard normal distributions):** To clarify, even though the weights of feature generating function (e.g. $\\omega$) are normally distributed, the random features themselves (i.e. $\\mathrm{ReLU}(\\omega^\\top x)$ are not, and thus our experiments make use of the generality of our theorem. Regardless, we will supplement Figure 1 and Figure 13 with $\\omega$ drawn from heavy tailed distributions.
> * **C9 (Invertibility assumption with ReLU activations):** We investigated other activations which fulfill the invertibility, including softplus (see e.g. Fig. 2 (left) and Fig. 9; note that using softplus activations, we can approximate ReLU activations arbitrarily precisely, see footnote 2 on page 4\)  and the Gaussian error function activation function (see e.g., Fig. 10 on the right hand side and Fig. 11 & 12), which satisfy the invertibility assumption.
> * **C11 (Double descent phenomena):** We are not entirely sure what aspect of the double descent phenomena you are referring to. If you clarify further, we would be happy to discuss it.
>
> **Responses to Questions:**
>
> - **Q1 (Training assumption):** Yes, in our theoretical analysis, we assume that the training procedure converges to the unique least norm solution, guaranteed for standard SGD initialized at zero. In our empirical analyses, we also always trained all models with the same training algorithms.
> - **Q2 (Epistemic vs. aleatoric uncertainty):** We admit that a single random feature model cannot decouple epistemic and aleatoric uncertainty. This decoupling is a possible advantage of ensembles, though our analysis in Section 3.3 implies that standard ensembles do not cleanly deliniate these sources either.
> - **Q3 (Feasibility of Assumption 2.1):** As discussed in the paper, the first condition is fulfilled whenever $w\_{\\perp i}$ and $w\_i^{\\top}$ are sub-Gaussian, which is true when the features come from activation functions with bounded derivatives and sub-Gaussian weights (which we would argue is a very weak assumption). For the second condition we expect that most nonlinear activation functions, such as sigmoid, tanh, sine, or cosine, satisfy this condition if we assume i.i.d. weights from a distribution with a density function. However, rigorously proving this is beyond the scope of this work.
> - **Q4 (Impact of regularization parameter $\\lambda$):** Intuitively, the training outcome is continuously dependent on the ridge parameter $\\lambda$; thus, small variations in $\\lambda$ typically result in minor deviations in predictions. The proof of our theorem effectively provides a rigorous confirmation for this intuition.
>
> We hope these responses and adjustments clarify our contributions and address your feedback. If you have further questions or suggestions, we would be happy to address them.

---

> > ### Comment · Reviewer_hF3L · 2025-04-03
> >
> > I thank the authors for integrating some comments in order to better clarify the work. I appreciate your efforts in answering my doubts and comments, which are convincing and clear. I therefore confirm to accept the paper.
> >
> > The only comment I would like to clarify is C11 about the double descent phenomena. This was not an important comment nor criticism. I just wanted to open the possibility to future consideration or debate. Since double descent is a phenomena in which overfitting is—counterintuitively—reversed with increasing model complexity (overparametrization) I thought that authors might have an interesting insight about it.
> >
> > The first reference about double descent can be read in (Nakkiran et al. 2021).
> >
> > Nakkiran, P., Kaplun, G., Bansal, Y., Yang, T., Barak, B., & Sutskever, I. (2021). Deep double descent: Where bigger models and more data hurt. *Journal of Statistical Mechanics: Theory and Experiment*, *2021*(12), 124003. https://arxiv.org/abs/1912.02292

---

### Official Review · Reviewer_CuSq · 2025-03-13

**Overall Recommendation:** 3

**Summary:**

This paper proves that in the random feature (RF) regression, the ensemble estimator is approximately equivalent with the simple regressor, as long as the model size is sufficiently great. This result can be applied not only to ridgeless models, but also to models with small ridge parameters.
Besides, the paper also demonstrate an interpretation of the predictive variance among ensemble members. It turns out that, in contrast with traditional models, the predictive variance of wide random feature ensembles quantifies the expected effects of increasing capacity rather than uncertainty.
These results implies that in this case, ensembles do not provide additional benefits over a simple model.

**Claims And Evidence:**

The statements of this paper are clear. A number of references and experimental data are provided to make the claims more convincing.
The main results of this paper warns that it is unwise to assume without theoretical guarantees the additional advantages of ensembles over simple models. In the background that more and more recent studies concentrate on random feature models, the contribution of this paper to our understanding of random feature learning is impressive and inspiring.

**Essential References Not Discussed:**

None.

**Experimental Designs Or Analyses:**

The details of experimental designs and results are clearly demonstrated in this paper.

**Methods And Evaluation Criteria:**

There are some limitations in this paper.
Firstly, there is some problem with Theorem 3.3. The bound term on the right hand side does not converge to zero as the number of feature $D$ increases, provided that $\delta$ is fixed. Thus, Theorem 3.3 is not sufficient to conclude that in the case of finite ensemble budget, the RF ensemble regressor and the RF simple regressor are asymptotically equivalent with high probability as $D$ goes to infinity.
On the other hand, Theorem 3.2 is impressive but not surprising. The linear random feature regression in the case of infinite width simply coincides with kernel interpolation, hence the infinite-width assumption together with the linearity of the model helps simplify the computations, as is shown in section 3.1 and appendix C.
Thus, we are concerned that these results and examples are not general and sophisticated enough.

**Other Comments Or Suggestions:**

None.

**Other Strengths And Weaknesses:**

None.

**Questions For Authors:**

Traditionally, many ensemble methods such as random forest are based on simple but non-linear method, and with the inspiration of this work, we are curious about what is the essential aspect of random feature model that prevent ensembles to bring extra advantages? Is it the linearity?

**Relation To Broader Scientific Literature:**

None.

**Theoretical Claims:**

We have checked the proofs and found no evident mistakes.

---

> ### Author Rebuttal · Authors · 2025-03-31
>
> We thank the reviewer for their comments and feedback on our paper. Below, we address the concerns and questions raised.
>
> **1\. Concerns about Theorem 3.3:**
>
> > "Firstly, there is some problem with Theorem 3.3. The bound term on the right-hand side does not converge to zero as the number of features increases, provided that $\\delta$ is fixed. Thus, Theorem 3.3 is not sufficient to conclude that in the case of finite ensemble budget, the RF ensemble regressor and the RF simple regressor are asymptotically equivalent with high probability as D goes to infinity."
>
> We believe there is a misunderstanding regarding this theorem. Theorem 3.3 does not aim to establish exact asymptotic equivalence. We proved asymptotic equivalence holding almost surely in our main result (Theorem 3.2). Thus, Theorem 3.3 is simply an analogous finite-sample guarantee showing that the single large model and the ensemble behave similarly with high probability at finite scales.
>
> **2\. Generality and novelty of Theorem 3.2:**
>
> > "On the other hand, Theorem 3.2 is impressive but not surprising. The linear random feature regression in the case of infinite width simply coincides with kernel interpolation, hence the infinite-width assumption together with the linearity of the model helps simplify the computations”
>
> We believe there is a misunderstanding regarding the key contribution of Theorem 3.2. The main contribution of the theorem is to investigate an *infinite* ensemble of *finite-width* random feature regressors rather than the infinite-width model. To our knowledge, no prior work has demonstrated that such an ensemble converges to the kernel interpolator, except in the special case of zero-mean Gaussian random features.
>
> **3\. Question about the essential aspect preventing ensemble advantages:**
>
> > "Traditionally, many ensemble methods such as random forest are based on simple but non-linear methods, and with the inspiration of this work, we are curious about what is the essential aspect of random feature model that prevents ensembles from bringing extra advantages? Is it the linearity?"
>
> As suggested by our title, the key attribute preventing ensembles of overparameterized random feature models from providing additional advantages is the **overparameterization** of the ensemble members. This claim is supported by our theoretical and empirical results contrasting ensembles of underparameterized and overparameterized random feature models (see Figure 2 and Appendix E) as well as other recent work (e.g., Abe et al., 2022).
>
> We hope these responses address your concerns and clarify the contributions of our work. If you have further questions or suggestions, we would be happy to address them.

---

> > ### Comment · Reviewer_CuSq · 2025-04-04
> >
> > Thank you for your clarification. I have increased my score.

---

### Decision · Program_Chairs · 2025-05-01

**Decision:**

Accept (oral)

**Comment:**

This paper offers a clear theoretical analysis showing that in overparameterized regimes, ensembles of random feature regressors provide minimal advantages compared to a single larger model, supported by thorough proofs and corroborating experiments. Reviewers commended the work’s strong theoretical base and its integration of relevant literature. Despite some concerns about finite-sample bounds, slight gaps between theory and practice, and potential expansions to additional activation functions, the authors’ rebuttal and clarifications were well received, and the empirical results on both synthetic and real data are sufficiently compelling. The paper’s insights into ensemble variance, its role in uncertainty quantification, and its demonstration that ensembling mainly replicates capacity expansion rather than offering robust uncertainty estimates advance our understanding of deep ensembles, which is important for ongoing work in kernel-based learning theories. Overall, the reviewers agreed that the paper’s contributions, especially its concrete analysis of how classical ensemble intuition may fail in highly overparameterized scenarios, are timely and valuable, and I recommend acceptance.